# EXPRESSIVE YET EFFICIENT FEATURE EXPANSION WITH ADAPTIVE CROSS-HADAMARD PRODUCTS

**Xuyang Zhang[1,2], Xi Zhang[2], Liang Chen[1,2][*] Hao Shi[1,2], Qingshan Guo[2]**
[1]Beijing Institute of Technology, Beijing, China
[2]Chongqing Innovation Center, Beijing Institute of Technology, Chongqing, China

## ABSTRACT

Recent theoretical advances reveal that the Hadamard product induces nonlinear representations and implicit high-dimensional mappings for the field of deep learning, yet their practical deployment in resource-constrained vision models remains largely unexplored. To address this gap, we introduce the Adaptive Cross-Hadamard (ACH) module, a novel operator that embeds learnability through differentiable discrete sampling and dynamic softsign normalization. This facilitates highly efficient feature reuse without incurring additional convolutional parameters, while ensuring stable gradient flow. Integrated into Hadaptive-Net (Hadamard Adaptive Network) via neural architecture search, our approach achieves unprecedented efficiency. Comprehensive experiments demonstrate state-of-the-art accuracy/speed trade-offs on image classification tasks, establishing Hadamard operations as specific building blocks for efficient vision models. The source code is available at `https://github.com/acelych/hadaptivenet`.

## 1 INTRODUCTION

Since AlexNet revolutionized computer vision (Krizhevsky et al., 2012), deep convolutional neural networks (CNNs) have advanced rapidly. Subsequent innovations mitigated gradient explosion via residual connections (He et al., 2016) and integrated self-attention into vision architectures (Dosovitskiy et al., 2020), gradually shifting model design toward greater depth for performance gains.

Conversely, lightweight networks (Howard et al., 2017; Zhang et al., 2018; Ma et al., 2018; Han et al., 2020) pursued efficiency. These models widely adopted the inverted bottleneck structure (e.g., MobileNets (Howard et al., 2017; Sandler et al., 2018; Howard et al., 2019; Qin et al., 2024), ConvNext (Liu et al., 2022; Woo et al., 2023)), which expands channel dimensions within blocks rather than compressing them. This design enables residual operations in lower-dimensional spaces, reducing computation while mitigating representational redundancy in high dimensions.

However, the inverted residual structure's dependency on repeated channel expansion/reduction operators inevitably introduces computational redundancy. Although effective, its dimension expansion phase requires significant convolution operations to project features into high-dimensional spaces, where substantial similarity exists across newly generated channels. GhostNet (Han et al., 2020; Tang et al., 2022; Liu et al., 2024) reveals this critical inefficiency, demonstrating that a large portion of expanded channels exhibit high linear correlations, and thus can be inexpensively synthesized via learned linear transformations of primary features rather than redundant convolutions. This breakthrough established the first generalized framework for feature reuse, bypassing costly dimension-specific operations. Subsequent works like FasterNet (Chen et al., 2023) further refined this paradigm, implementing feature reuse via partial convolution operators that selectively merge spatially neighboring features.

Our work revisits feature recombination efficiency from an orthogonal perspective: instead of generating or filtering features, we exploit the intrinsic nonlinear representational capacity of learnable Hadamard products to achieve ultra-efficient feature fusion. The Hadamard product (a.k.a. element-wise multiplication), as a highly practical method, has long garnered significant attention in the fields

---

[*]Corresponding author. (`chenl@bit.edu.cn`)

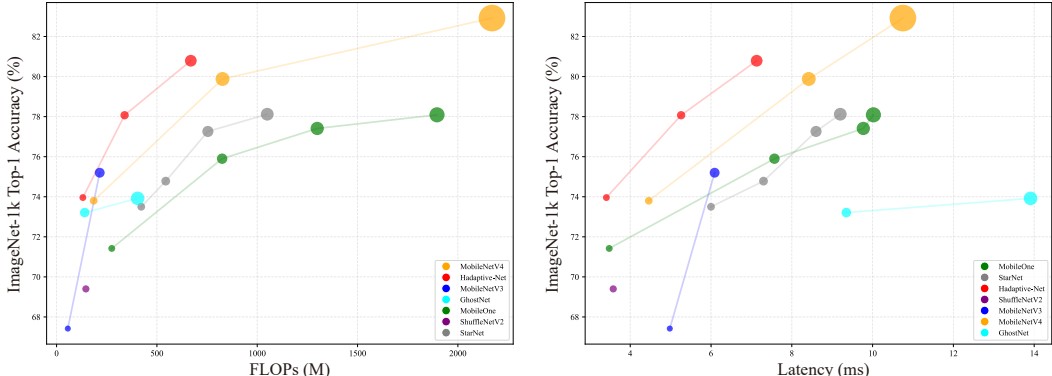

Figure 1: **The trade-off between FLOPs/latency and top-1 accuracy.** These diagrams compare the efficiency among different state-of-the-art models with ours Hadaptive-Net in image classification task. Detailed experimental configurations are provided in section 5.3.

of deep learning. Recently, it became a new learning paradigm in the field of lightweight network design owing to effective performance and concise computation. Its principle is straightforward, for two identical matrices $\mathbf{A}, \mathbf{B}$:

$$\mathbf{C} = \mathbf{A} \odot \mathbf{B} \Leftrightarrow C_{i,j} = A_{i,j} \cdot B_{i,j}$$

Recent theoretical advances reveal that stacked Hadamard products can induce nonlinear representations and implicitly high-dimensional mappings when deeply cascaded (Ma et al., 2024). Capitalizing on these insights, we propose the Adaptive Cross-Hadamard (ACH) module. This novel operator transcends conventional Hadamard usage by embedding learnability through two key mechanisms: (i) channel attention-guided feature gating, and (ii) differentiable discrete sampling. Thus, ACH establishes Hadamard products as foundational deep learning operators while enabling parameter-free feature reuse.

To effectively deploy the ACH module, we construct Hadaptive-Net (Hadamard Adaptive Network) through differentiable neural architecture search (NAS), jointly optimizing model topology and ACH integration points. For efficient on-device execution, we further develop tailored GPU acceleration strategies addressing computation scheduling challenges. In comparative experiments, Hadaptive-Net outperforms state-of-the-art efficient models, achieving higher accuracy with lower computational costs (fig. 1).

## 2 RELATED WORK

This section reviews two types of previous studies related to this work: the application of Hadamard product and efficient model design.

### 2.1 RESEARCHES IN HADAMARD

It can be learned from Chrysos et al. (2025) that the taxonomy for applying the Hadamard product in deep learning is divided into four categories: high-order interactions, multimodal fusion, adaptive modulation, and efficient operators. Ma et al. (2024) and Chen et al. (2022a) reveal its ability to implicitly induce high-order nonlinear mappings. As an example of multimodal fusion, Kim et al. (2017) uses the Hadamard product to achieve low-rank bilinear pooling as an approximation of full bilinear pooling. Adaptive modulation—also referred to as the gating mechanism, such as in LSTMs (Hochreiter & Schmidhuber, 1997) —is a widely adopted application of the Hadamard product. For instance, HAda (Wang et al., 2024) employs it to scale weights generated by a hypernetwork in multi-view learning scenarios, while HiRA (Huang et al., 2025) applies it to construct high-rank weight updates during the fine-tuning of large language models. MogaNet (Li et al., 2024) also uses the Hadamard product to adaptively focus on informative features by fusing multi-scale depthwise separable convolutions with varying dilation rates.

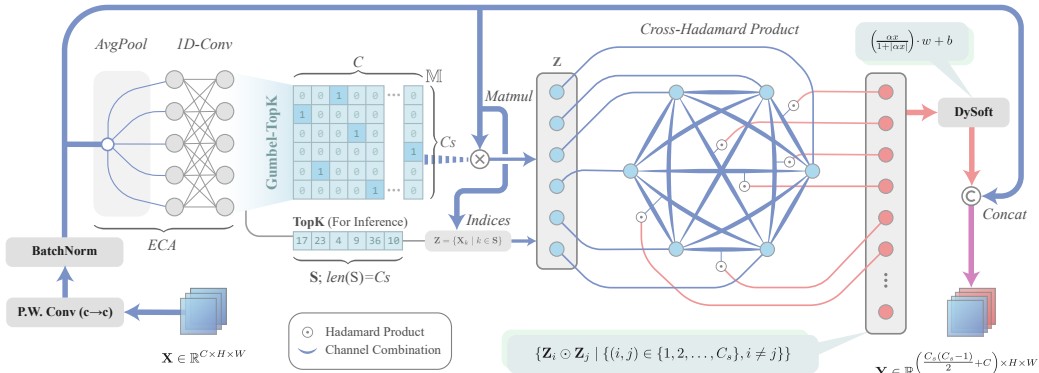

Figure 2: **Illustration of the ACH module.** Input features **X** undergo linear transformation and batch normalization. An ECA module generates channel-wise scores, with Gumbel-Topk sampling (training) or top-k selection (inference) determining active channels. Selected features **Z** undergo cross-Hadamard product, normalized by dynamic softsign, then concatenated with original features.

The forms of efficient operators are quite diverse. To mitigate the $\mathcal{O}(n^2)$ complexity of Transformers, some approaches replace matrix multiplications in attention mechanisms with Hadamard products, as seen in FocalNet (Yang et al., 2022) and HorNet (Rao et al., 2022). Gu & Dao (2023) and Zhu et al. (2024) adopt the Hadamard product as a core operator for ultra-efficient feature expansion and nonlinear fusion via channel-wise cross-products. However, existing methods suffer from critical limitations: fixed combination rules (inter- or intra-channel) restrict optimization flexibility, and predefined operations limit interpretability. We therefore propose enhancing Hadamard products with learnable channel expansion capabilities, transforming them into dedicated deep learning operators that leverage inherent nonlinearity while overcoming previous rigidity.

## 2.2 EFFICIENT MODEL DESIGN

The pursuit of efficient architectures has driven continuous innovation: from SqueezeNet's pioneering use of pointwise convolutions (Iandola et al., 2016), to MobileNetV1's depthwise separables (Howard et al., 2017), MobileNetV2's inverted bottlenecks (Sandler et al., 2018), and ShuffleNet's channel shuffling (Zhang et al., 2018; Ma et al., 2018). Neural Architecture Search (NAS) further advanced efficiency in MnasNet (Tan et al., 2019), EfficientNet (Tan & Le, 2019), and MobileNetV3 (Howard et al., 2019), culminating in MobileNetV4's universal inverted bottlenecks (Qin et al., 2024). Concurrently, vision transformers inspired hybrid designs like Mobile-Former (Chen et al., 2022b) and EdgeViT (Pan et al., 2022).

Feature reuse mechanisms provide complementary efficiency: GhostNet revealed channel-wise redundancies in conventional convolutions, replacing redundant features via linear transformations (Han et al., 2020; Tang et al., 2022). FasterNet constrained convolution ranges (Chen et al., 2023), while GhostNetV3 (Liu et al., 2024) and MobileOne (Vasu et al., 2022) adopted RepVGG's reparameterization (Ding et al., 2021) to merge parallel branches.

## 3 METHODOLOGY

This section establishes a hierarchical framework for the ACH module, progressing from mathematical foundations to architectural deployment. First, we formalize the Hadamard product's role in channel expansion. Second, we introduce differentiable discrete sampling via Gumbel-TopK with adaptive temperature annealing, enabling end-to-end channel selection. Third, to stabilize dynamically generated features, DySoft normalization replaces statistical normalization with bounded sigmoidal activation. Finally, we integrate ACH into Hadaptive-Net through gradient-based NAS.

### 3.1 Hadamard for Channel Expansion

Inspired by the properties of high-dimensional mapping and non-linearity, we observe that the Hadamard product aligns well with the characteristic of neural networks that gradually increase channel dimensions while reducing spatial dimensions. This suggests that the Hadamard product is particularly suitable for channel expansion.

Specifically, we compute the Hadamard product for pairwise combinations of input channels while retaining the original feature maps. This can be expressed as:

$$\mathbf{Y} = \mathbf{X} \oplus \{\mathbf{X}_i \odot \mathbf{X}_j \mid \{(i,j) \in \{1, 2, \ldots, C\}, i \neq j\}\}$$
$$\text{s.t.} \quad \mathbf{X} \in \mathbb{R}^{C \times H \times W}, \mathbf{Y} \in \mathbb{R}^{\frac{C(C+1)}{2} \times H \times W} \tag{1}$$

where $\mathbf{X}$ represents the input feature map, $\mathbf{X_i}$ and $\mathbf{X_j}$ denote the $i$-th and $j$-th channels of $\mathbf{X}$, $\odot$ denotes Hadamard product, and $\oplus$ denotes channel-wise concatenation, respectively. This approach can be seen as putting the initial features $\mathbf{X}$ and the features after transformation into the same feature space. More specifically, the stitched feature vector can be understood as a high-dimensional vector, and the original feature space can be regarded as a set of bases, providing interpretability for the composite features that carry implicit high-dimensional information.

Based on these insights, we designed the Adaptive Cross-Hadamard module, which is illustrated as fig. 2. The design details and learnable methods of the module will be discussed in the following sections.

### 3.2 Differentiable Discrete Sampling

As feature maps propagate through deep networks, their channel dimensions expand dramatically, causing the number of possible channel interactions to grow quadratically. This combinatorial explosion makes exhaustive pairwise computation prohibitively expensive. Even for modest channel counts, practical implementations require selecting a fixed subset of channels for efficient processing. We thus reformulate eq. (1) as:

$$\mathbf{Y} = \mathbf{X} \oplus \{\mathbf{Z}_i \odot \mathbf{Z}_j \mid \{(i,j) \in \{1, 2, \ldots, C_{(s)}\}, i \neq j\}\}$$
$$\text{s.t.} \quad \mathbf{X} \in \mathbb{R}^{C \times H \times W}, \mathbf{Y} \in \mathbb{R}^{\left(\frac{C_{(s)}(C_{(s)}-1)}{2} + C\right) \times H \times W} \tag{2}$$
$$\text{s.t.} \quad \mathbf{Z} = \{\mathbf{X}_k \mid k \in \mathbf{S}\}$$

where $\mathbf{S}$ represents a sequence of chosen channels' indexes and $C_{(s)}$ indicates the amount of chosen channels.

However, this selection process is inherently discrete, posing a challenge for gradient-based optimization. Thus, we introduced Gumbel-Topk trick (Gumbel, 1954) for selecting procedure. Formally, we donate scores of each channels as a vector $\xi$, which is obtain from an ECA module (Wang et al., 2020):

$$\xi = \text{ECA}(\mathbf{X}) = \mathcal{P}(\mathbf{X}) * W + b \tag{3}$$

where $\mathcal{P}$ denotes adaptive average pooling, $*$ denotes a 1D convolution operation. Then calculate the probability distribution as below:

$$\mathbf{M}_c = \frac{\exp\left(\frac{\xi_c + o_c}{\tau}\right)}{\sum_{c'=1}^{C} \exp\left(\frac{\xi_{c'} + o_{c'}}{\tau}\right)} \quad c \in C \tag{4}$$
$$\text{s.t.} \quad o_i = -\log(-\log(u)), \ u \sim \text{Unif}\,[0, 1]$$

where $o_i$ are i.i.d sampled from Gumbel distribution, $\mathbf{M}$ denotes a probability distribution vector resulted from softmax, and $\tau$ denotes temperature parameter that controls the smoothness of the softmax output, respectively. The Gumbel-distributed perturbations $o_i$ inject controlled stochasticity into the discrete selection process, ensuring channels temporarily still receive gradient feedback. This prevents over-reliance on initial channel selections while maintaining alignment with the ECA's distribution across forward passes. The temperature parameter $\tau$ governs output sharpness: higher values yield softer selections, while $\tau \to 0$ produces one-hot behavior.

While $\mathbf{M}$ is continuous and differentiable, it leads to a discrete and nondifferentiable vector $\mathbf{M^H}$. With straight through estimator (STE) technique (Bengio, 2013):

$$\mathbf{M}_c^H = I_{\mathbf{S}}(c), \quad \mathbf{S} = \text{top-k}(\mathbf{M}, k = C_{(s)}) \quad c \in [1, C] \quad \text{forward} \tag{5}$$

$$\frac{\partial \mathcal{L}}{\partial \xi} := \frac{\partial \mathcal{L}}{\partial \mathbf{M^H}} \cdot \frac{\partial \mathbf{M}}{\partial \xi} = \frac{\partial \mathcal{L}}{\partial \mathbf{M^H}} \cdot \frac{\partial \text{softmax}(\xi/\tau)}{\partial \xi} \quad \text{backward} \tag{6}$$

where $I_A(x)$ denotes the indicator function, discrete $\mathbf{M^H}$ could conduct data stream during training and gradient could skip through $\mathbf{M^H}$ to $\mathbf{M}$ during backpropagation. Since hyperparameter $\tau$ modulates the intensity of continuous values influence the selection of discrete values through softmax, Consequently, the adjustment of $\tau$ should be responsive to gradient variations. Instead of relying on a global parameter scheduler, the ACH module employs adapting $\tau$ dynamically based on the norm of historical gradients, thereby preserving the end-to-end training characteristics:

$$\begin{aligned} \tau &\leftarrow \text{CLAMP}\left(\tau \cdot (1 + \alpha \cdot \text{sign}(\|grad\|_2 - \tau_{hist})), \, 0.01, \, 4.0\right) \\ \tau_{hist} &\leftarrow \|grad\|_2 \end{aligned} \tag{7}$$

This design specifically addresses layer-wise heterogeneity through dynamic responsiveness: $\tau$ increases when current gradient norms exceed historical values (enhancing exploration for diverse features), while decreasing when gradients diminish (accelerating semantic-specific convergence). Refer to appendix A.1 for the detailed procedure applied during each training epoch. To continuously maintain gradient propagation, following steps require matrix operations:

$$\mathbb{M}'_{s,c} = \delta(c, \mathbf{S}_s) \quad \forall s \in [1, C_{(s)}], c \in [1, C] \tag{8}$$

$$\mathbb{M}_s = \mathbb{M}'_s \odot \mathbf{M^H} \quad s \in C_{(s)} \tag{9}$$

where $\mathbb{M}$ denotes a one-hot mapping matrix from input channels to selected channels, $\delta(a, b)$ denotes Kronecker delta function. Given eq. (2) and eq. (9), we can finally obtain $\mathbf{Y}$ in eq. (2) with gradient computation graph:

$$\begin{aligned} \mathbf{Y} &= \mathbf{X} \oplus \{\mathbf{Z}_i \odot \mathbf{Z}_j \mid \{(i,j) \in \{1, 2, \ldots, C_{(s)}\}, i \neq j\}\} \\ \text{s.t.} \quad \mathbf{Z} &= \mathbb{M} \cdot \mathbf{X} \end{aligned} \tag{10}$$

For inference stage, it directly takes the first few bits of the output of the ECA module and uses this as the index to extract the channels that need to be calculated, saving unnecessary calculation.

### 3.3 DYSOFT NORMALIZATION

The cross-Hadamard product creates input-adaptive channel combinations that enhance nonlinearity but produce unstable output distributions. Unlike conventional convolutions that rely on statistical normalization, this dynamic behavior renders batch normalization (Ioffe & Szegedy, 2015) ineffective and risks gradient explosion. Inspired by recent success of activation-based normalization in Transformers (Zhu et al., 2025), we propose DySoft, a dynamic softsign normalization that intrinsically bounds outputs while maintaining hardware efficiency:

$$y = \frac{\alpha x}{1 + |\alpha x|} \cdot w + b \tag{11}$$

Table 1: **Comparison of dynamic sigmoidal curves.** The experimental conditions are the experimental results of replacing the normalized layers of all cross Hadamard products of the small model finally determined in section 5.

|  | Sigmoid | Softsign | Alge. Sigmoid |
|---|---|---|---|
| Formula | $\frac{e^x}{e^x+1}$ | $\frac{x}{1+|x|}$ | $\frac{x}{\sqrt{1+x^2}}$ |
| Top1(%) | 73.14 | **73.57** | 72.80 |

Table 2: **Performance Comparison of ACH Module Replacement on MobileNetV3.** There are a total of 11 Inverted Bottleneck modules in the network, with indices starting from 0 in the table. Several modules were selected for the ablation experiment. The first row of the table represents the replaced layer(s), and the second row represents the Top1 accuracy (%). '/' denotes the original unmodified MobileNetV3-S, 'IB' denotes Inverted Bottleneck.

| / | $\underline{IB^0}$ | $IB^1$ | $IB^9$ | $\underline{IB^8}$ | $IB^{10}$ | $IB^{9,10}$ |
|---|---|---|---|---|---|---|
| 70.01 | 69.74 | 69.74 | 69.89 | 70.38 | 71.03 | **71.58** |

Table 3: **Neural Architecture Search Confidence Distribution.** Showing selection confidence between Ghost and ACH variants. Underlined channels indicate downsampling layers. Fixed layers (no search) marked with hyphens. Values below 0.01% are indicated as <0.01%. DySig, DyAlge represent dynamic sigmoid and algebraic sigmoid, as the abbreviation DySoft, respectively.

| Channels | Ghost Conf. | ACH Variants Conf. | | |
|---|---|---|---|---|
|  |  | DySoft | DySig | DyAlge |
| $\underline{32}$ | - | - | - | - |
| $\underline{48}$ | - | - | - | - |
| 32 | - | - | - | - |
| 64 | **99.97%** | <0.01% | <0.01% | <0.01% |
| 64 | **99.99%** | <0.01% | <0.01% | <0.01% |
| 96 | **99.87%** | <0.01% | <0.01% | <0.01% |
| 96 | **99.69%** | 0.24% | <0.01% | <0.01% |
| 96 | **99.48%** | 0.18% | 0.05% | 0.23% |
| 96 | **98.40%** | 0.80% | 0.15% | 0.51% |
| 96 | **97.37%** | 0.63% | 0.63% | 1.10% |
| 96 | 2.92% | **62.02%** | 6.58% | 28.18% |
| $\underline{128}$ | 12.78% | **43.72%** | 23.87% | 19.62% |
| 128 | 0.34% | **70.05%** | 25.34% | 4.24% |
| 128 | 1.68% | **34.56%** | 29.40% | 34.30% |
| 960 | - | - | - | - |

where $\alpha, w, b$ denote learnable factors of an affine transform. Empirical comparisons table 1 show softsign outperforms tanh and algebraic sigmoid variants in stability and computational efficiency, making it ideal for mobile deployment. The indispensability of DySoft is discussed in appendix A.2.

### 3.4 HADAMARD ADAPTIVE NETWORK

To systematically validate the efficacy, implementability and architectural compatibility of the proposed ACH module, we construct Hadaptive-Net (Hadamard Adaptive Network), a network family that serves as a testbed for ACH module. We employ gradient-based Neural Architecture Search (NAS) (Dong & Yang, 2019) not to produce a single, static architecture, but as a principled methodology to discover the optimal integration of ACH modules within a modern, efficient backbone. This approach allows us to objectively evaluate ACH's performance and unearth general design principles for its deployment, mitigating the biases of manual heuristic design.

Our search is informed by a preliminary analysis revealing that ACH is depth-dependent, performing best in late-stage layers (table 2). We thus designed a search space co-integrating ACH with GhostNet-style modules, enabling NAS to select the optimal operator per layer. The search results (table 3) confirm our hypothesis: ACH is preferentially selected over Ghost modules in high-dimensional spaces. The finalized Hadaptive-Net architectures are derived from these discovered principles, with full specifications in the appendix A.3.

### 4 IMPLEMENTATION

Prior to additional experimentation, we must ensure the cross-Hadamard product, as a novel operator, attains its theoretical efficiency on CPU/GPU and other hardware.

### 4.1 COMPUTATIONAL COMPLEXITY ANALYSIS

The computational complexities for expanding channel dimension with a feature map of $f \times f$ from $m$ to $n$ dimensions with $k \times k$ convolution are analyzed as below. For inverted bottleneck:

$$\mathcal{O}(mn \cdot f^2)_{\text{pointwise conv}} \tag{12}$$

For Ghost module, which partially replaces the expensive pointwise convolution with a more efficient strategy:

$$\mathcal{O}(ms \cdot f^2)_{\text{pointwise conv}} + \mathcal{O}((n-s) \cdot k^2 f^2)_{\text{cheap op}} \qquad (13)$$

Our method preserves the pointwise convolution while delegating channel expansion to Hadamard product operations:

$$\mathcal{O}(m^2 \cdot f^2)_{\text{pointwise conv}} + \mathcal{O}((n-m) \cdot f^2)_{\text{hadamard}} \qquad (14)$$

Since $m \ll n$, The computational complexity of Ghost module is reduced to $\frac{s}{n}$ of inverted bottleneck convolution, while our ACH module achieves approximately $\frac{1}{m}$ of the inverted bottleneck convolution's complexity in channel expansion. Derivations are shown in appendix A.5. Remarkably, each Hadamard-derived feature map requires only $f^2$ FLOPs, achieving superior efficiency compared to conventional approaches.

The emphasis on FLOPs over latency is driven by the necessity to maintain cross-platform compatibility and approximate theoretical performance limits. While such a prioritization can be readily implemented and validated in serial processing architectures, heterogeneous computing systems present significant challenges that necessitate extensive optimization efforts.

## 4.2 GPU ACCELERATION

While lower theoretical computational complexity typically suggests faster inference speed, the actual GPU execution involves intricate scheduling by the CPU. The sophisticated channel mapping process in ACH module often gets decomposed into multiple sub-operations by inference frameworks, manifesting as frequent CPU-GPU synchronization and repeated kernel launches. The triangular computation pattern of $C_n^2$ combinations for cross-Hadamard products necessitates specialized operator design, for which we propose two optimization approaches:

1. **Direct-Indexing**: Each thread block exclusively handles one Hadamard product. The closed-form mapping from pairing index $p$ to to matrix indices $(i, j)$ is:

$$\begin{cases} i = \frac{1}{2} \cdot \lfloor (2n-1) - \sqrt{(2n-1)^2 - 8p} \rfloor \\ j = i + 1 + p - \frac{i \cdot (2n-1-i)}{2} \end{cases}$$

   where $n$ denotes number of candidate channels.

2. **Parity-Balanced**: Assign c thread blocks (c: input channels), evenly distributing irregular computations via iterative indexing algorithm 2, then compute pairing indices with an inverse formula:

$$p = \frac{i \cdot (2n-i-1)}{2} + (j-i-1)$$

Table 4 demonstrates the acceleration effects of different optimization approaches on the Hadaptive-Net-L, which confirms the indispensability of optimization in the step of implementation. To approach ex-

Table 4: Acceleration performance.

| | (Native) | Direct-Indexing | Parity-Balanced |
|---|---|---|---|
| Latency (ms) | 12.40 | 7.21 | 7.13 |

treme performance, we analyzed the performance of the two algorithms under different characteristic scales. Details of the experiments and Parity-Balanced algorithm are shown in appendix A.5. Resulting in practical inference scenarios, it's recommended to employ the parity-balanced approach for high-channel/small-HW tensors, while considering direct-indexing for spatial dimensions near 32× multiples. For performance-critical applications, custom compilation of tilling strategies matching factors of specific spatial dimensions may be warranted.

## 5 EXPERIMENT

This section demonstrates the applied scenarios of our proposed method. All experiments on ACH module are modified from the configuration of image classification experiment. All latency benchmarks were conducted within the ONNX Runtime (developers, 2021) framework. To demonstrate

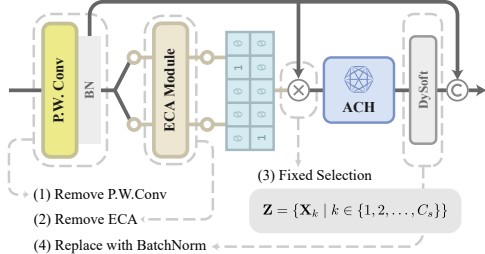

| P.W.Conv | ECA | Learnable | DySoft | Top-1 |
|:---:|:---:|:---:|:---:|:---:|
| ✓ | | | ✓ | 69.27 |
| ✓ | | ✓ | ✓ | 69.12 |
| | ✓ | ✓ | ✓ | 71.96 |
| ✓ | ✓ | ✓ | | 64.39 |
| ✓ | ✓ | ✓ | ✓ | **73.57** |

Figure 3: **Component-wise ablation.** Illustration of component-wise ablation variations with component-accuracy table. (1) and (2) represent removal of pointwise convolution and ECA module, respectively. (3) represents the replacement of learnable selection with fixed channel combinations, and (4) represents the substitution of cross-Hadamard normalization with standard batch normalization.

Table 5: **Replacements of ACH module on efficient models.** We replace the last two layers of each model. For instance, replacing last two universal inverted bottleneck modules for MobileNetV4.

Table 6: **Performance of Hadaptive-Net on object detection.** We employ the SSD object detector to different scales of Hadaptive-Net and baseline models with COCO (Lin et al., 2014) dataset.

| Model | Top-1 (%) | Params (M) | FLOPs (M) |
|---|---|---|---|
| MobileNetV3-S | 70.01 | 1.61 | 123 |
| MobileNetV3-S (repl.) | 71.58↑ | 1.55↓ | 114↓ |
| MobileNetV4-S | 73.15 | 2.62 | 385 |
| MobileNetV4-S (repl.) | 72.19↓ | 2.98↑ | 381↓ |
| ShuffleNetV2-1.0 | 65.89 | 1.36 | 303 |
| ShuffleNetV2-1.0 (repl.) | 71.68↑ | 1.28↓ | 291↓ |
| StarNet-S1 | 71.84 | 2.68 | 854 |
| StarNet-S1 (repl.) | 72.07↑ | 2.56↓ | 810↓ |

| Backbone | mAP@0.5:0.95 | mIOU |
|---|---|---|
| MobileNetV3-S | 21.7 | 71.2 |
| MobileNetV2-1.0 | 21.9 | 70.5 |
| GhostNetV3-1.0 | 22.7 | 72.8 |
| Hadaptive-Net-S | 22.1 | 72.4 |
| Hadaptive-Net-M | 22.9 | 73.0 |
| Hadaptive-Net-L | **23.2** | **73.4** |

the practical optimization potential of our proposed ACH module, we implemented it as a custom CUDA operator. We stress that this result is presented to showcase the high optimizability of the ACH operator. It should not be interpreted as a strict, head-to-head speed comparison against baseline models, which utilize the standard, framework-provided operators native to ONNX Runtime.

## 5.1 ABLATION ON ACH MODULE

This experiment evaluates the contribution of each ACH component through controlled ablations: (1) Whether to keep pointwise convolution layer. (2) Whether to keep ECA module. (3) Learnable selection or fixed combinations. (4) Dynamic softsign or batch normalization. The baseline model of this set of experiments is obtained from the best model of the previous set of experiments. Fig. 3 illustrates the variations of the ablation experiments with presenting the quantitative results, revealing that all the components serve their respective functions. The pointwise convolution provides fundamental channel-wise information exchange, while the ECA module enables the assessment of channel importance. These two components establish the essential foundation for the module's learnability. Disabling this learnability nearly renders the module ineffective, demonstrating that the discrete differentiation mechanism can properly provide gradients for the former components.The employment of dynamic softsign effectively circumvents gradient explosion risks, consequently exhibiting markedly better performance than batch normalization in experimental trials.

## 5.2 PLUG-AND-PLAY VERSATILITY OF ACH MODULE

The ACH module's distinct mechanism enhances semantic feature representation, making it ideal for standalone integration. We validate this by replacing the final two layers of four state-of-the-art effi-

Table 7: **Comparison of efficient models.** This table presents parameter counts, computational complexity (FLOPs), and latency measurements obtained from the CIFAR-100 (Krizhevsky, 2009) dataset.

| Model | Params (M) | FLOPs (M) | Latency GPU (ms) | Latency CPU (ms) | Latency Mobile (ms) | Top-1 Accuracy CIFAR-100 (%) | Top-1 Accuracy ImageNet-1k (%) |
|---|---|---|---|---|---|---|---|
| MobileNetV3-S (Howard et al., 2019) | 1.62 | 56 | 4.98 | 33.21 | 6.71 | 70.01 | 67.42 |
| MobileOne-S0 (Vasu et al., 2022) | 2.10 | 275 | 3.48 | 30.70 | 5.11 | 69.70 | 71.42 |
| Hadaptive-Net-S (ours) | 2.10 | 131 | 3.41 | 29.45 | 4.28 | **73.57** | **73.96** |
| ShuffleNetV2-1.0 (Ma et al., 2018) | 2.28 | 146 | 3.58 | 40.60 | 4.55 | 65.89 | 69.40 |
| MobileNetV4-S (Qin et al., 2024) | 2.62 | 185 | 4.46 | 24.68 | 4.31 | 73.15 | 73.80 |
| StarNet-S1 (Ma et al., 2024) | 2.68 | 422 | 6.00 | 82.73 | 7.96 | 71.84 | 73.50 |
| Hadaptive-Net-M (ours) | 3.09 | 339 | 5.26 | 39.81 | 6.47 | **74.10** | **78.07** |
| StarNet-S2 (Ma et al., 2024) | 3.43 | 544 | 7.30 | 94.50 | 8.41 | 67.70 | 74.78 |
| GhostNet-1.0 (Han et al., 2020) | 4.03 | 140 | 9.35 | 87.59 | 10.02 | 72.01 | 73.21 |
| MobileNetV3-L (Howard et al., 2019) | 4.33 | 215 | 6.09 | 56.41 | 6.66 | 72.81 | 75.20 |
| MobileOne-S1 (Vasu et al., 2022) | 4.82 | 825 | 7.57 | 40.76 | 7.98 | 72.97 | 75.90 |
| StarNet-S3 (Ma et al., 2024) | 5.49 | 754 | 8.60 | 112.7 | 9.87 | 68.27 | 77.26 |
| Hadaptive-Net-L (ours) | 6.11 | 669 | 7.13 | 57.62 | 9.11 | **74.73** | 80.79 |
| StarNet-S4 (Ma et al., 2024) | 7.22 | 1050 | 9.20 | 134.0 | 12.24 | 68.97 | 78.12 |
| MobileOne-S2 (Vasu et al., 2022) | 7.80 | 1299 | 9.77 | 61.19 | 10.23 | 73.25 | 77.41 |
| GhostNetV3-1.0 (Liu et al., 2024) | 8.13 | 404 | 13.91 | 180.54 | 19.07 | 73.20 | 73.92 |
| MobileNetV4-M (Qin et al., 2024) | 8.56 | 827 | 8.42 | 47.93 | 9.36 | 74.66 | 79.88 |
| MobileOne-S3 (Vasu et al., 2022) | 10.15 | 1896 | 10.02 | 81.28 | 10.36 | 73.80 | 78.09 |
| MobileNetV4-L (Qin et al., 2024) | 31.44 | 2170 | 10.75 | 79.94 | 11.71 | 74.38 | **82.92** |

cient networks: MobileNetV3 (Howard et al., 2019), MobileNetV4 (Qin et al., 2024), ShuffleNetV2 (Ma et al., 2018), and StarNet (Ma et al., 2024), with ACH. As table 5 shows ACH improves accuracy in all networks except MobileNetV4 while reducing computational costs, confirming its generalizability as a plug-and-play performance enhancer. See appendix A.6 for deeper analysis.

## 5.3 IMAGE CLASSIFICATION

We evaluate the performance of Hadaptive-Net on image classification (CIFAR-100 (Krizhevsky, 2009), ImageNet-1K (Deng et al., 2009)), conducting comprehensive comparisons with other state-of-the-art efficient models. Our experiments use PyTorch with AdamW optimizer (lr=0.001, momentum=0.9, weight decay=1e-4) and CrossEntropyLoss. Training employs cosine annealing with 5% linear warmup over 200 epochs (batch=64, 224×224 inputs). We conducted the experiments both on CIFAR-100 (Krizhevsky, 2009) and ImageNet-1K (Deng et al., 2009). Latency tests use ONNX-converted models (batch=1), 500-run average (Hardware details in appendix A.4).

**Result**: According to table 7, Hadaptive-Net achieves superior accuracy in the first two groups while maintaining relatively low computational requirements. Although MobileNetV4 demonstrates the best performance in the largest parameter group, this comes at the cost of significantly higher computational overhead.

## 5.4 OBJECT DETECTION

To validate the generalization capability of HadaptiveNet as a backbone network across different downstream tasks, we conduct object detection experiments using the SSD (Liu et al., 2015) framework. All models are trained on COCO train2017 (Lin et al., 2014) with a fixed input resolution of 320×320 for 120 epochs, employing synchronized SGD optimization (momentum=0.9, weight decay=5e-4) and cosine learning rate decay initialized at 0.02. The detection head utilizes focal loss ($\gamma$ =2.0) for classification and smooth L1 loss for bounding box regression. Evaluation follows the standard COCO protocol reporting mAP@[0.5:0.95] on val2017. For implementation details see appendix A.4.

**Result**: As shown in table 6, Hadaptive-Net continues the high-level performance of image classification tasks in the extended task of object detection. This proves that Hadaptive-Net has a more general feature extraction ability.

**Justification**: Maximizing the efficacy of the ACH module likely requires an end-to-end co-design. We believe the more profound opportunity presented by this study lies in leveraging the principles of structured, lightweight cross-channel interaction embodied by ACH to redesign bottleneck components like Feature Pyramid Networks (FPNs), focusing on efficiently fusing multi-scale feature information. This represents a highly promising direction for breaking the efficiency bottleneck of current detectors.

## 5.5 GENERALITY ON TRANSFORMER

The Multi-Head Self-Attention (MHSA) mechanism in Transformer (Vaswani et al., 2017) focuses on the N-dimension, i.e., the relationships between tokens, while the Feed-Forward Network (FFN) operates on the C-dimension, integrating semantic information carried and aggregated within individual tokens. The FFN typically follows a classic inverted bottleneck structure, where the ACH module can effectively play a role in computational compression.

To maintain research consistency, focusing on computer vision tasks and lightweight design, we have decided to supplement our experiments with improvements on the MobileViT (Mehta & Rastegari, 2022) model. Specifically, we replaced the FFN layers of the middle four Transformer encoders in MobileViTs with the ACH module.

**Result**: The comparative results are shown in table 8, which demonstrate that replacing half of the FFN layers with the ACH module yielded significant improvements. Notably, this enhancement was achieved without increasing the number of parameters or computational complexity (FLOPs), leading to su-

Table 8: Replacements of ACH module on MobileViTs.

| Model | Params(M) | GFLOPs | Top1-Acc | Top5-Acc |
|---|---|---|---|---|
| MobileViT-small | 4.55 | 2.879 | 71.70 | 92.16 |
| (Replaced) | 4.40 | 2.822 | 72.42 | 92.20 |
| MobileViT-x-small | 1.80 | 1.559 | 69.98 | 91.43 |
| (Replaced) | 1.74 | 1.537 | 70.48 | 91.71 |
| MobileViT-xx-small | 0.88 | 0.588 | 67.62 | 90.19 |
| (Replaced) | 0.85 | 0.579 | 67.42 | 90.32 |

perior performance on the CIFAR-100 dataset compared to the baseline. This experiment substantiates that the ACH module exhibits a promising level of generalizability within deep learning, particularly for the role of a channel feature extractor. For natural language processing related attempts, please refer to appendix A.6.

## 6 CONCLUSION

This work systematically transforms Hadamard products from auxiliary operations into specific deep learning primitives, culminating in the development of the novel Adaptive Cross-Hadamard (ACH) module and its integration into Hadaptive-Net. Theoretical and empirical analyses show ACH's superiority over depthwise separable convolutions in computational efficiency and representational capacity. Lastly, this work establishes Hadamard-based operations as a valuable direction for efficient deep learning architectures, offers insights for integrating novel mathematical operations into neural network design.

## REPRODUCIBILITY STATEMENT

For theoretical verification, refer to appendix A.5 for computational complexity analysis. For implementation reproducibility, section 4 discusses the whole principle of engineering acceleration algorithm kits. For training details, appendix A.3 shows the training configuration of hyperparameters and hardware set. The code to implement the module and models in this paper has been open source.

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

# A APPENDIX

## A.1 TRAINING MECHANISM

$\tau$ **Adjustment**: We implement distinct temperature control mechanisms for ACH modules versus NAS due to fundamental differences in their training paradigms. For ACH modules distributed across network layers, which process heterogeneous features and semantics, we deliberately design a adaptive regulation algorithm based on gradient norm trends:

---

**Algorithm 1** $\tau$ Adjustment via Gradient Norm Tracking

---

**Input**: Current gradient tensor $grad$, scaling factor $\alpha = 0.01$
**Parameter**: Historical gradient norm $\tau_{hist}$, current temperature $\tau$
**Output**: Updated temperature $\tau$

1: **if** $\tau_{hist} \neq 0 \wedge grad \neq$ NULL **then**
2:    $\Delta \leftarrow \begin{cases} 1 & \text{if } \|grad\|_2 \geq \tau_{hist} \\ -1 & \text{otherwise} \end{cases}$
3:    $\tau_{new} \leftarrow \tau \cdot (1 + \alpha \cdot \Delta)$
4:    $\tau \leftarrow \text{CLAMP}(\tau_{new}, 0.01, 4.0)$
5: **end if**
6: **if** $grad \neq$ NULL **then**
7:    $\tau_{hist} \leftarrow \|grad\|_2$
8: **end if**
9: **return** $\tau$

---

For differentiable NAS (GDAS), which operates under fundamentally different optimization constraints, we retain the GDAS framework's global annealing strategy:

- Training protocol: Architecture parameters undergo periodic updates separate from model weights, with gradient clipping ($\|\nabla\| \leq 1.0$) ensuring stable convergence.

- Temperature scheduler: Implements predefined decay strategies:

$$\tau_e = \begin{cases} \tau_{\max} - (\tau_{\max} - \tau_{\min}) \cdot \frac{e}{E} \\ \tau_{\max} \cdot \left(\frac{\tau_{\min}}{\tau_{\max}}\right)^{\frac{e}{E}} \\ \tau_{\min} + 0.5(\tau_{\max} - \tau_{\min}) \cdot \left[1 + \cos\left(\pi \frac{e}{E}\right)\right] \end{cases}$$

The three formulas represent linear, exponential, cosine annealing, respectively. Consistent with GDAS methodology, $\tau$ remains within $[0.1, 4.0]$ throughout training.

**NAS Specifics**: The neural architecture search process employs a dual-optimizer framework with distinct settings for model parameters and architecture parameters. Training executes over 250 epochs with a global batch size of 64, utilizing CUDA acceleration on a single GPU device (ID 0). The primary model optimizer is AdamW with base learning rate 0.003, momentum 0.9, and weight decay $1e-4$, coupled with CosineAnnealingLR scheduling for learning rate decay. Architecture parameters undergo separate optimization via AdamW with specialized learning rate $3e-4$ to accommodate their distinct gradient distributions.

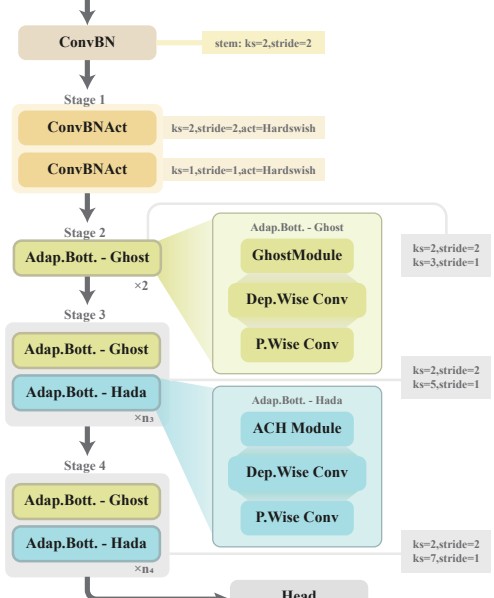

Figure 4: Hadaptive-Net architecture overview.

Table 9: Hadaptive-Net-S architecture details.

| Layer | Module | Arguments |
| --- | --- | --- |
| 0 | CNA | [3, 32, 2, 2] {BN, None} |
| 1 | CNA | [32, 48, 2, 2] {BN, HS} |
| 2 | CNA | [48, 32, 1, 1] {BN, HS} |
| 3 | AB | [32, 64, 'Ghost', 4.0, 2, 2] |
| 4 | AB | [64, 64, 'Ghost', 2.0, 3, 1] |
| 5 | AB | [64, 96, 'Ghost', 4.0, 2, 2] |
| 6 | AB | [96, 96, 'Hada', 16, 5, 1] |
| 7 | AB | [96, 96, 'Hada', 16, 5, 1] |
| 8 | AB | [96, 96, 'Ghost', 2.0, 5, 1] |
| 9 | AB | [96, 96, 'Hada', 16, 5, 1] |
| 10 | AB | [96, 96, 'Hada', 16, 5, 1] |
| 11 | AB | [96, 128, 'Ghost', 6.0, 2, 2] |
| 12 | AB | [128, 128, 'Hada', 32, 7, 1] |
| 13 | AB | [128, 128, 'Hada', 32, 7, 1] |
| 14 | CNA | [128, 960, 1, 1] {BN, HS} |
| 15 | FN | [960, 100, 1280, 0.3] |

## A.2 DYSOFT INDISPENSABILITY

Quantitative experiments can not explain the indispensability of DySoft since the model without DySoft training is extremely unstable and has no representative experimental data. However, these phenomena can illustrate a problem that DySoft is empirically necessary, which could be explained by probability theory.

**Problem Tracing**: Traditional normalization methods, such as BatchNorm (Ioffe & Szegedy, 2015) and LayerNorm (Ba et al., 2016), have a priori assumption that the statistical mean and statistical variance of the tensors they receive are knowable and traceable, which constitutes the basis of model convergence. In the process of ACH training and reasoning, we will involve a standard $Z_i \odot Z_j$ cross Hadamard product calculation. In previous machine learning methods, the use of Hadamard product is usually self referential, that is, $Z^2 = Z \odot Z$. In this case, we can easily infer the mean value of $Z^2 = Z \odot Z$ from the mean and variance $\mu, \sigma^2$ of $Z$ :

$$\text{Var}(Z) = \text{E}[(Z - \mu)^2] = \text{E}[Z^2] - (\text{E}[Z])^2$$
$$E[Z^2] = \mu^2 + \sigma^2$$

Since tensor $Z$ was processed by normalization from above layer, which approximately satisfies $Z \sim N(\mu, \sigma^2)$. According to the fourth moment formula of normal distribution:

$$
\begin{aligned}
\mathrm{E}[Z^4] &= \mu^4 + 6\mu^2\sigma^2 + 3\sigma^4 \\
\mathrm{Var}(Z^2) &= \mathrm{E}[Z^4] - (\mathrm{E}[Z^2])^2 \\
&= (\mu^4 + 6\mu^2\sigma^2 + 3\sigma^4) - (\mu^2 + \sigma^2)^2 \\
&= 2\sigma^2(2\mu^2 + \sigma^2)
\end{aligned}
$$

If the self referring Hadamard product is deformed, for example $\phi_1(Z) \odot \phi_2(Z)$, Let $\phi$ here be a linear transformation operator, the corresponding matrix form is $X_1, X_2$ ($X \in \mathbb{R}^{m \times n}$), bias vectors are $b_1, b_2$ ($b \in \mathbb{R}^m$), then:

$$
\mathrm{E}[\phi(Z)] = \mathrm{E}[XZ + b] = \mu \cdot \frac{\sum_i^m \sum_j^n X_{i,j}}{m} + \mathrm{E}[b]
$$

For variance, since $Z$ can approximate normal distribution, here we assume that its elements are i.i.d, then there are:

$$
\mathrm{Var}(\phi(Z)) = \frac{1}{m} \cdot \sum_i^m \mathrm{Var}(\phi(Z)_i) = \frac{1}{m} \cdot \sum_i^m \sum_j^n A_{i,j}^2 \cdot \sigma^2 = \sigma^2 \cdot \frac{\|A\|_F^2}{m}
$$

Suppose $\phi$ is a nonlinear transformation operator, which does not directly exist the predictability of analytical solutions. However, the purpose of normalization method is not to accurately track the statistical representation of tensors, but to ensure that the statistical representation of tensors remains stable in the reasoning process.

Let the mapping $T_f : \mathbb{R} \times \mathbb{R}_{>0} \to \mathbb{R} \times \mathbb{R}_{\geq 0}$ be: $(\mu, \sigma^2) \mapsto (\mu', \sigma'^2)$. If $T_f$ unbounded, that is, there is a sequence $(\mu_k, \sigma_k^2)$ such that $\|T_f(\mu_k, \sigma_k^2)\| \to \infty$ as long as a layer accidentally reaches the state (such as disturbance, initialization deviation), the next layer of statistics will be unstable; If $T_f$ is discontinuous or the derivative is unbounded (e.g. $f(z) = 1_{z>0}$ is at $\mu = 0$), small disturbance can lead to $\mu', \sigma'^2$ upheaval, resulting in unstable training.

BatchNorm is generally considered in CV tasks. BN independently estimates the mean and variance of $k$ for each channel:

$$
\hat{\mu}_k = \mathbb{E}_{\mathrm{x} \sim \mathcal{B}}[x_k], \quad \hat{\sigma}_k^2 = \mathrm{Var}_{\mathrm{x} \sim \mathcal{B}}(x_k)
$$

And perform channel by channel affine transformation:

$$
x_k' = \gamma_k \cdot \frac{x_k - \hat{\mu}_k}{\sqrt{\sigma_k^2 + \epsilon}} + \beta_k
$$

This operation does not force statistical consistency between channels, but allows or even encourages significant statistical heterogeneity between channels:

$$
\exists\, i \neq j \quad \text{s.t.} \quad \hat{\mu}_i \neq \hat{\mu}_j,\ \hat{\sigma}_i^2 \neq \hat{\sigma}_j^2
$$

This property is consistent with the inductive bias of "channel division" in convolutional networks - different channels can professionally respond to different semantic patterns (such as edge, texture, color), which is the key basis for its high representation efficiency in visual tasks. In contrast, LN is normalized in the sample dimension:

$$
\mathrm{x}' = \gamma \cdot \frac{\mathrm{x} - \mu}{\sigma} + \beta, \quad \mu = \frac{1}{C} \sum_k^C x_k,\ \sigma^2 = \frac{1}{C} \sum_k^C (x_k - \mu)^2
$$

The implicit priori is that all channels at the same spatial location should have the same statistical scale, which drives statistical convergence between channels. This assumption is compatible with the inductive bias of "all tokens are comparable" in the global attention mechanism (such as ViT),

but in CNN dominated by local receptive fields, it will weaken the channel specific characterization ability and lead to performance degradation.

Let us consider $y_{ij} = x_i \odot x_j$, its output statistics depend on the joint second moment of the input channel. Under the heterogeneity distribution induced by BN, let $x_i \sim \mathcal{N}(\mu_i, \sigma_i^2)$, $x_j \sim \mathcal{N}(\mu_j, \sigma_j^2)$ and i.i.d, then:

$$\mathbb{E}[y_{ij}] = \mu_i \mu_j \mathrm{Var}(y_{ij}) = \mu_i^2 \sigma_j^2 + \mu_j^2 \sigma_i^2 + \sigma_i^2 \sigma_j^2$$

When the channel statistics differ significantly (e.g. $|\mu_i| \gg |\mu_j|$ or $\sigma_i \gg \sigma_j$), the variance shows a multiplicative amplification effect, which is far beyond the single channel scale range. The affine parameters of BN are only channel specific, which can not effectively correct the new statistical offset caused by such cross-channel coupling. Otherwise, the pairing process of $i, j$ is obtained by the nonlinear transformation of each input, which makes it impossible for the statistical representation iterative map $T_f$ to find the fixed point.

Although LN normalization may be used inside ACH module, it is very important to understand the heterogeneity between channels in CV tasks. There is usually a typical CNN trunk containing BN upstream of the module, so the whole feature learning process has been dominated by the heterogeneity of BN a priori. The model's understanding of image semantics will evolve spontaneously towards the direction of "channel specialization". At this time, if a strong cross-channel nonlinear module with implicit homogeneity assumption is inserted into the reasoning chain, it will lead to a priori conflict.

**Solution**: The DySoft we introduced is essentially a variant of the softsign activation function:

$$y = \frac{\alpha x}{1 + |\alpha x|} \cdot w + b, \quad \lim_{\alpha x \to \pm\infty} \frac{\alpha x}{1 + |\alpha x|} = \pm 1$$

Due to the boundedness of softsign, no matter how large the input variance $\sigma^2$ is, the output variance is rigidly limited in the $(0, 1)$ range; When the input is small, it shows approximate linearity and maintains the characteristics of the signal. The parameter $\alpha$ can dynamically balance the expression and compression of the layer.

When the cross-Hadamard $y_{ij} = x_i \odot x_j$ has variance like $\sigma_i, \sigma_j \gg 1$, $\mathrm{Var}(y_{ij})$ increased by $\mathcal{O}(\sigma^4)$. After accessing DySoft, this trend can be significantly compressed and given boundedness. In addition, DySoft is also designed based on the hypothesis of channel heterogeneity, which is a priori compatible with the heterogeneity of BN. Its $w, b$ parameters are channel specific, and can independently learn the scale and offset for each cross-Hadamard product channel. At the same time, it does not destroy the channel professional representation established by the upstream BN, and only makes local intervention on the "danger signal", thus realizing the organic unity of characterization and stability.

In summary, DySoft is a learnable statistical compression gating (SCG) module, which achieves hard variance clamping for high square error input through bounded nonlinear mapping $\mathcal{S}(u) = u/(1 + |u|)$ and restores the characterization capacity in combination with channel specific affine transformation. Without violating the heterogeneity prior of batch normalization, the design effectively inhibits the growth of multiplicative variance caused by cross channel nonlinearity (such as cross Hadamard product), and makes the statistical map $T : (\mu, \sigma^2) \mapsto (\mu', \sigma'^2)$ bounded and smooth, so as to meet the core condition of "knowability", providing a stable and convergent statistical target for the normalization layer.

### A.3 HADAPTIVE-NET CONFIGURATION

This section mainly shows the results of three groups of NAS experiments and the decision of final Hadaptive-Net structure. See tables 10 to 12 for NAS experiments details.

Hadaptive-Net adopts a hierarchical backbone architecture comprising a stem followed by four distinct stages, as shown in fig. 4. To implement Ghost and ACH module with adaptability, we design the Adaptive Bottleneck that can decide the expansion layer of the bottleneck manually. The net-

Table 10: **Neural Architecture Search Result (a).** Compared with different kernel sizes. Reaching 67.55% top1-acc as result.

| Channels | Ghost Conf. | | ACH Conf. | | | Blank |
|---|---|---|---|---|---|---|
| | 2 | 3 | 3 | 5 | 7 | |
| 32 | - | - | - | - | | |
| 48 | - | - | - | - | | |
| 32 | - | - | - | - | | |
| 64 | 54% | 46% | - | - | - | - |
| 64 | - | 24% | - | 76% | - | - |
| 96 | 52% | 48% | - | - | - | - |
| 96 | 27% | 14% | 9% | 50% | - | - |
| 96 | 25% | 21% | 28% | 26% | - | - |
| 96 | 20% | 17% | 23% | 17% | - | 23% |
| 96 | 19% | 19% | 23% | 17% | - | 23% |
| 96 | 20% | 20% | 20% | 18% | - | 22% |
| 128 | 37% | 28% | 17% | 17% | - | - |
| 128 | - | - | 35% | 33% | 33% | - |
| 128 | - | - | 60% | 27% | 13% | - |
| 128 | - | - | 1% | 1% | 1% | 96% |
| 128 | - | - | 1% | 1% | 1% | 96% |
| 128 | - | - | 1% | 2% | 1% | 96% |
| 960 | - | - | - | - | - | - |

Table 11: **Neural Architecture Search Result (c).** Shows the distribution of ACH configurations across different channel sizes. Values represent percentage confidence (rounded to nearest integer). '-' indicates layers that were not searched. Format: ACH-[chosen_dim]-[kernel_size]. Reaching 67.73% top1-acc as result.

| Channels | ACH Conf. | | | |
|---|---|---|---|---|
| | 16-3 | 16-5 | 32-3 | 48-3 |
| 32 | - | - | - | - |
| 48 | - | - | - | - |
| 32 | - | - | - | - |
| 64 | - | - | - | - |
| 64 | - | - | - | - |
| 96 | - | - | - | - |
| 96 | 68% | 12% | 20% | - |
| 96 | 44% | 13% | 43% | - |
| 96 | 47% | 16% | 37% | - |
| 96 | 35% | 20% | 45% | - |
| 96 | 45% | 15% | 39% | - |
| 128 | - | - | - | - |
| 128 | 19% | - | 52% | 29% |
| 128 | 28% | - | 32% | 40% |
| 128 | 36% | - | 41% | 22% |
| 128 | 40% | - | 30% | 30% |
| 128 | 51% | - | 23% | 26% |
| 960 | - | - | - | - |

Table 12: **Neural Architecture Search Result (b).** Shows the distribution of ACH configurations across different channel sizes. Values represent percentage confidence (rounded to nearest integer). '-' indicates layers that were not searched. Since training ACH modules requires a lot of iterations to be effective, the network tends to skip them during training. Reaching 66.23% top1-acc as result.

| Channels | ACH Conf. | | | Blank |
|---|---|---|---|---|
| | 16 | 32 | 48 | |
| 32 | - | - | - | - |
| 48 | - | - | - | - |
| 32 | - | - | - | - |
| 64 | - | - | - | - |
| 64 | - | - | - | - |
| 96 | - | - | - | - |
| 96 | 14% | 9% | - | 68% |
| 96 | 14% | 13% | - | 65% |
| 96 | 16% | 13% | - | 62% |
| 96 | 13% | 15% | - | 61% |
| 96 | 13% | 14% | - | 60% |
| 128 | - | - | - | - |
| 128 | 1% | 1% | 1% | 98% |
| 128 | 1% | 1% | 1% | 98% |
| 128 | 1% | 1% | 1% | 98% |
| 128 | 1% | 1% | 1% | 98% |
| 128 | 1% | 1% | 1% | 98% |
| 960 | - | - | - | - |

work begins with a linear convolutional layer as the stem, followed by fixed two conventional convolutional layers in Stage 1 for initial feature extraction. Stage 2 incorporates two fixed Adaptive Bottlenecks utilizing Ghost module as expansion layers, enabling rapid downsampling. Stages 3 and 4 employ Ghost Ada.Bott. for downsampling layers and Hadamard Ada.Bott for repeated residual blocks, with particular emphasis on parameter concentration in Stage 3, following ConvNeXt's

design philosophy. The kernel sizes progressively increase across stages, with non-downsampling layers configured as $1 \times 1$, $3 \times 3$, $5 \times 5$, and $7 \times 7$ respectively.

Refer to table 9 for detailed description of layer level architecture configuration. CNA denotes combinition of convolution, normalization and activation layers. AB denotes adaptive bottleneck, which could be subdivided into Ghost module or ACH (Adaptive Cross-Hadamard) module. Hada denotes the ACH module. BN denotes batch normalization. HS denotes hardswish activation. FN denotes full connection layer. All first two arguments represent input/output channel. All last two arguments represents kernel size and stride size, respectively.

## A.4 EXPERIMENTS DETAILS

The following is a detailed description of the experimental part of this paper.

**Hardware Configurations**: Latency tests conducted on:

- **Desktop GPU**: NVIDIA RTX TITAN (24GB GDDR6, CUDA 11.6)
- **Server CPU**: Intel Xeon Gold 5218 (2.3GHz, 16C/32T)
- **Mobile SoC**: Qualcomm Snapdragon 870 (4×Cortex-A77@2.4GHz + 1×A77@3.2GHz, Adreno 650)

All tests used ONNX Runtime 1.16.0 with default execution providers.

**Object Detection - Training Protocol**: The base learning rate of 0.02 corresponds to a batch size of 64 distributed across 5 GPUs, scaled linearly according to the batch size. We apply 3-epoch linear warmup and reduce the learning rate to 1e-5 via cosine scheduling. Data augmentation includes random HSV color jittering with hue delta limited to 18 degrees and saturation scaling between 0.5-1.5, followed by random canvas expansion up to 2x original size and IoU-based cropping with thresholds sampled from [0.1,0.3,0.5].

**Object Detection - Architecture Specifications** The SSD detector generates 6 default boxes per feature map location with aspect ratios spanning [1:1, 1:2, 1:3, 2:1, 3:1]. Feature maps are extracted from five backbone stages with strides of [8,16,32,64,128] pixels respectively, corresponding to spatial dimensions from 38×38 down to 1×1. During focal loss computation we set the $\alpha$-balancing parameter to 0.25 after empirical validation across the range [0.1,0.5].

## A.5 IMPLEMENTATION DETAILS

This section will supplement the derivation of previous computational complexity analysis and implementation details of GPU acceleration algorithm mentioned in the original text.

With eqs. (12) to (14), we can derive the ratio of Ghost Module complexity to that of the standard pointwise convolution:

$$\text{Ratio}_{\text{Ghost}} = \frac{m \cdot s \cdot f^2 + (n - s) \cdot k^2 \cdot f^2}{m \cdot n \cdot f^2} = \frac{s}{n} + \frac{n - s}{n} + \frac{k^2}{m} \tag{15}$$

Since $s$ is often chosen as a fraction of $n$ (e.g., $s = n/2$), the term $\frac{n-s}{n}$ is approximately a constant (e.g., $1/2$). Since $k^2$ is small (e.g., 9 for a $3 \times 3$ kernel) and m can be relatively large, the term $\frac{k^2}{m}$ is often negligible. Under the condition $m \ll n$, the simplified complexity ratio is:

$$\text{Ratio}_{\text{Ghost}} \approx \frac{s}{n} \tag{16}$$

Similarly, we calculate the efficiency ratio by comparing ACH module complexity to the standard pointwise convolution:

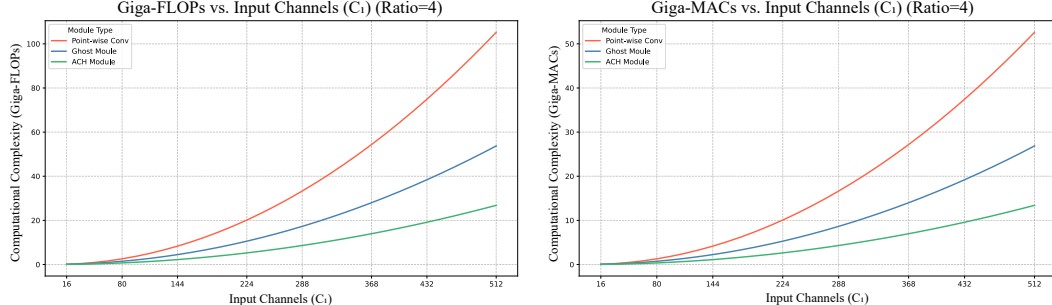

Figure 5: Comparison of computational efficiency under different input channel sizes

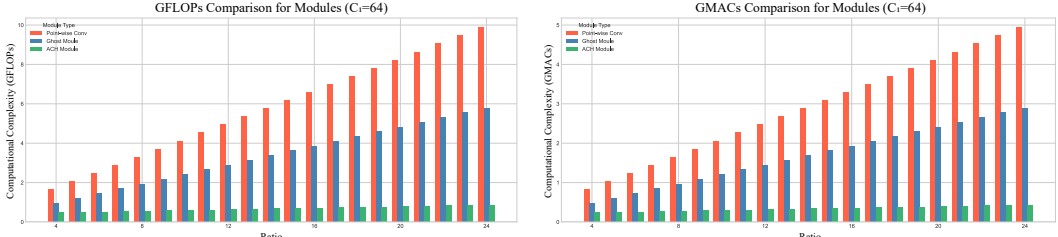

Figure 6: Comparison of computational efficiency under different expansion ratios

$$\text{Ratio}_{\text{ACH}} = \frac{m^2 \cdot f^2 + (n-m) \cdot f^2}{m \cdot n \cdot f^2} = \frac{m^2 + n - m}{m \cdot n} = \frac{m}{n} + \frac{1}{m} + \frac{1}{n} \tag{17}$$

Given the constraint $m < n, 1 \ll m, 1 \ll n$, the terms $\frac{1}{m}$ and $\frac{1}{n}$ become very small and can be considered negligible. Thus, the simplified complexity ratio for the ACH module is:

$$\text{Ratio}_{\text{ACH}} \approx \frac{m}{n} \tag{18}$$

For this theoretical analysis, we have carried out several groups of measured data on different input channel sizes and amplification ratios to confirm that ACH module has a very strong reasoning speed compared with the standard point-by-point convolution and ghost module.

The experiment is carried out for two specific situations: fixed 4-fold scaling ratio, 16-512 different input channel sizes; Fixed 64 input channel size, 4-24 times scaling ratio (224*224 per frame). We counted the Multiply-ACCumulate Operations (MACs) and Floating Point Operations (FLOPs) of the two groups of experiments as illustrated in figs. 5 and 6.

This ratio can be shown intuitively in the above experimental results. From the experimental results, the computational complexity ratio under different channel sizes is relatively fixed, while different scaling ratios, which is $m/n$, show a linear relationship.

Previous section 4.2 presented two GPU acceleration algorithms for ACH operators. One algorithm, the Direct-Indexing, is implemented as the name suggests. Another algorithm, the Parity-Balanced, could be written as algorithm 2.

To systematically evaluate these methods under varying tensor configurations (batch/channel dimensions versus spatial sizes), we conducted comparative experiments using square matrices (same sized height & width). See fig. 7 for the experiment details and results.

Both algorithms demonstrate relatively stable performance across varying batch sizes, indicating comparable parallelism in channel-agnostic scenarios. Despite both are expanding dimensionality, the parity-balanced approach exhibits superior optimization for high-channel tasks compared to high-batch scenarios, owing to its specialized load balancing for channel-dense tensors.

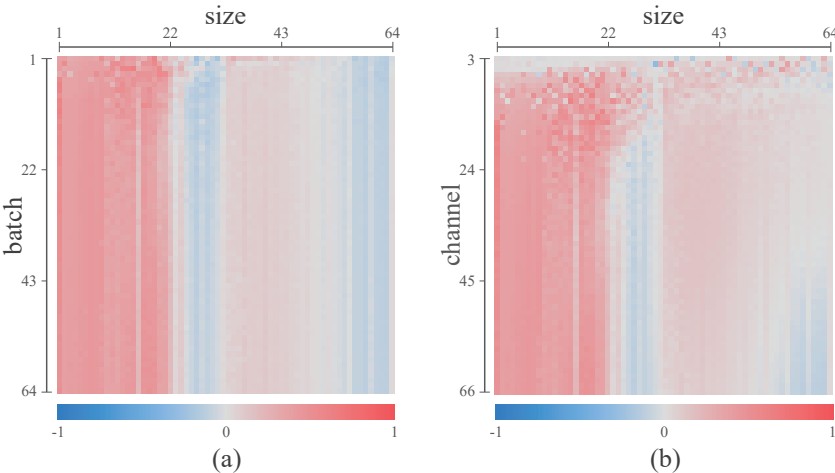

Figure 7: **Normalized difference heatmap of optimization approaches runtime.** Color-coded visualization of relative performance between direct-indexing ($A$) and parity-balanced ($B$) approaches using $\frac{A-B}{A+B+\epsilon}$, where red indicates A is slower (B more efficient) and blue indicates the opposite. (a) Batch size versus spatial dimensions scaling. (b) Channel count versus spatial dimensions scaling.

For feature maps with smaller spatial dimensions, the parity-balanced approach significantly outperforms direct-indexing due to: (1) The balanced approach's input tensor reuse pattern enhances L1/L2 cache hit rates in GPU global memory, reducing memory access latency while increasing arithmetic intensity per thread block through reduced thread block maintenance. (2) While appearing to introduce serialization, the balanced method effectively concentrates inevitable serial processes within individual thread blocks, as GPU core counts cannot simultaneously satisfy all computational demands for dimensionally dense small tensors, thereby avoiding context-switching overhead. (3) Direct-indexing requires separate thread block allocation per matrix computation, leading to underutilized warp resources when small matrices cannot fill the thread block size.

When spatial dimensions approach integer multiples of 32 (thread block dimension), direct-indexing prevails due to thread blocks achieve near-saturation load conditions with peak artificial intensity, and the method's end-to-end processing better aligns with hardware scheduling optimizations.

---

**Algorithm 2** Parity-Balanced Indexing Strategy

---

**Input**: Channel count $c$
**Parameter**: Thread block group ID $id$
**Output**: Choosen channels $i, j$

1: **for** $it \leftarrow 0$ to $c - 1$ **do**
2:     **if** $it < id \wedge \neg((id - it) \bmod 2)$ **then**
3:         $i \leftarrow it, j \leftarrow id$
4:     **else if** $it > id \wedge (id - it) \bmod 2$ **then**
5:         $i \leftarrow id, j \leftarrow it$
6:     **else**
7:         **continue**
8:     **end if**
9:     Compute Hadamard product for matrices $i$ and $j$
10: **end for**

---

### A.6 EXTENDED EXPERIMENTS

**Grad-CAM**: To further elucidate the role of the ACH module in enhancing the model's representational capacity, we designed two sets of comparative experiments using Grad-CAM visualization

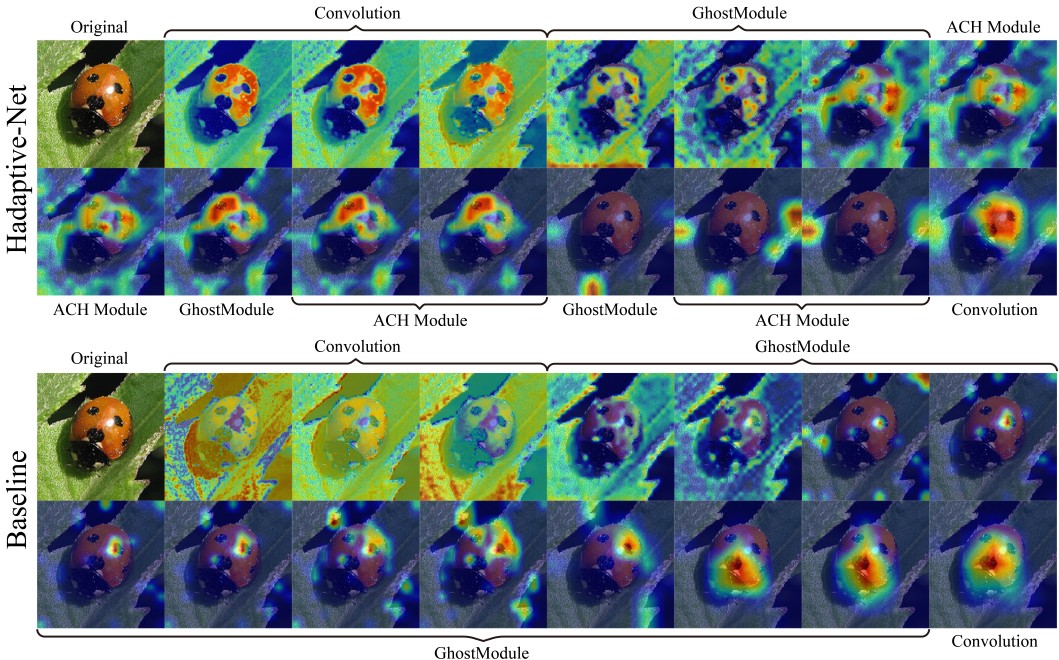

Figure 8: **Network visualization via Grad-CAM across layers (1).** Simple scenario: ladybug. Downward arrows denote downsampling layers.

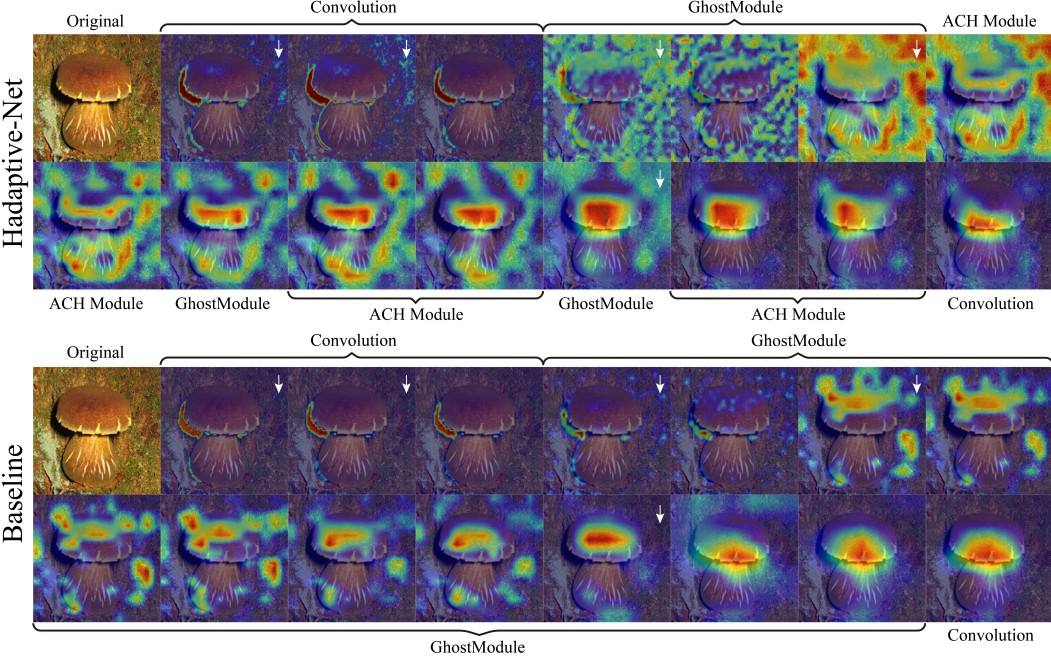

Figure 9: **Network visualization via Grad-CAM across layers (2).** Complex scenario: mushroom. Downward arrows denote downsampling layers.

to examine the changes brought by the ACH module compared to a conventional convolutional network. For clearer and more intuitive comparison, we adopted as the baseline a modified version of Hadaptive-Net-S in which all ACH modules were replaced with Ghost modules, in order to demonstrate the feature extraction pattern under purely linear transformations.

Table 13: Replacements of ACH module on BERT.

| Layers | Modify | In-Out-Channel | Mid-Channel | SST-2 | MNLI-10% |
|--------|--------|----------------|-------------|-------|----------|
| BERT-6 | FFN (fixed channel) | 764 | 764 | 64.7 | 36.7 |
| BERT-6 | ACH Module | 764 | 2043 | 76.2 | 41.8 |
| BERT-6 | FFN | 764 | 2048 | 80.3 | 46.7 |
| BERT-12 | FFN (fixed channel) | 764 | 764 | 53.1 | 34.3 |
| BERT-12 | ACH Module | 764 | 2043 | 75.9 | 41.2 |
| BERT-12 | FFN | 764 | 2048 | 80.6 | 45.5 |

The first experiment, which is shown as fig. 8, involves a simple scenario, where a ladybug is clearly distinguishable from the background. The baseline model exhibits a standard processing pattern that progresses from texture analysis to focal emphasis. In contrast, Hadaptive-Net not only extracts texture more accurately, but also achieves target focus with fewer layers, while performing more precise edge segmentation. After the final downsampling step, the baseline model continues attempting to focus on the main subject, whereas Hadaptive-Net begins to attend to the edges of withered leaves, suggesting an attempt to capture higher-level semantic correlations.

The second experiment, which is shown as fig. 9, presents a more complex situation, where a mushroom exhibits some color overlap with the background. Compared to the baseline, Hadaptive-Net transitions more rapidly from the edge extraction phase to the target focusing phase, and explores a larger spatial area, indicating a larger effective receptive field.

In summary, the introduction of the ACH module not only reduces computational complexity but also endows the model with more powerful semantic representation capabilities.

**NLP Attempt**: We tried to extend ACH module to NLP. We conducted a comparative test on the 6-layers and 12-layers BERT (Devlin et al., 2019). The accuracy of the BERT model using different channel feature extractors (FFN, ACH module and standard FFN with unchanged middle layer dimension) was tested in SST-2 (Socher et al., 2013) and 10% MNLI datasets (Williams et al., 2018).

The models were evaluated on two standard natural language understanding benchmarks: the Stanford Sentiment Treebank (SST-2) for binary sentiment classification and the Multi-Genre Natural Language Inference (MNLI) dataset for textual entailment. For SST-2, the model was trained and evaluated on the full dataset. To assess performance in a low-resource setting, the model was trained on a 10% stratified subset of the MNLI training set and evaluated on the full matched validation set.

The training configuration was consistent across both tasks. Models were trained for 3 epochs with a global batch size of 32 and evaluated with a batch size of 64. The optimization used a learning rate of 2e-5 with a linear warmup over the first 10% of the training steps and weight decay of 0.01. The models, which followed a BERT-base architecture (12 layers, 12 attention heads, 768-dimensional hidden states), were initialized with random weights. Input sequences were tokenized using the 'bert-base-uncased' tokenizer with a maximum length of 128 tokens. The sole evaluation metric was classification accuracy, calculated as the percentage of correctly predicted labels against the ground truth. All experiments were run on a single GPU without mixed-precision training.

As a result in table 13, the performance of ACH module was not stunning enough to exceed the BERT baseline. However, from the perspective of the motivation of compressing the calculation scale, ACH module still plays a big role. Only adding a small amount of cross-Hadamard product in one step can approach the FFN without channel depth to a better level, revealing its potential as a unique algorithm in the field of NLP.

As we are mainly engaged in CV related work and lack relevant experience in NLP field, the experimental setup may be a little rough. If there are any problems, readers are welcome to correct them.

## B  THE USE OF LARGE LANGUAGE MODELS

In the preparation of this work, the author(s) utilized a Large Language Model (LLM) primarily to aid in polishing and refining the writing. The tool was used for purposes such as improving gram-

matical correctness, enhancing sentence fluency, and rephrasing for clarity. All ideas, theoretical analyses, experimental designs, results, and conclusions remain entirely those of the author(s). The final manuscript has been thoroughly reviewed and edited by the author(s), who take full responsibility for all content presented herein.

