# OpenReview forum: "Expressive yet Efficient Feature Expansion with Adaptive Cross-Hadamard Products"
_ICLR.cc/2026/Conference — ICLR 2026 Poster_

### Official Review · Reviewer_3mhu · 2025-10-15

**Soundness:** 3
**Presentation:** 3
**Contribution:** 3
**Rating:** 6
**Confidence:** 2

**Summary:**

This paper introduces the Adaptive Cross-Hadamard (ACH) module, an operator that embeds learnability through differentiable discrete sampling and dynamic softsign normalization. The ACH module is designed to enable parameter-free feature reuse while stabilizing gradient propagation. It is integrated into the Hadaptive-Net (Hadamard Adaptive Network) using neural architecture search to achieve high efficiency.

**Strengths:**

(1) The paper's approach of leveraging the Cross-Hadamard Product from the perspective of feature expansion to enhance model efficiency is novel and interesting.

(2) The paper is well-written, providing a comprehensive description from its methodology to the implementation details.

**Weaknesses:**

(1) Insufficient Motivation for Technical Choices: The paper lacks a clear and in-depth discussion of the motivation behind its core technical components. For instance, in the sections on "HADAMARD FOR CHANNEL EXPANSION" and "DIFFERENTIABLE DISCRETE SAMPLING," the paper focuses more on the implementation details (how) rather than the fundamental reasons (why) these specific techniques are suitable or advantageous. This makes it difficult for readers to fully appreciate the design principles of the proposed method.

(2) Potentially Limited Novelty in Sub-components: The novelty of some technical components appears to be incremental. The section on "DIFFERENTIABLE DISCRETE SAMPLING," for example, seems to rely heavily on the combination of existing techniques like the Gumbel-Topk trick and the ECA module. While combining existing ideas can be a valid contribution, the paper does not sufficiently articulate what makes this specific combination novel or non-trivial.

(3) Limited Experimental Scope: The experimental evaluation is confined to CNN-based architectures. In the current deep learning landscape, demonstrating effectiveness on Transformer-based models is crucial for showing broad applicability, especially for a method focused on efficiency. The absence of such experiments makes the generalizability of the proposed Hadaptive-Net unclear.

**Questions:**

(1) In the sections on "HADAMARD FOR CHANNEL EXPANSION" and "DIFFERENTIABLE DISCRETE SAMPLING," the paper primarily describes how these techniques are applied but lacks a detailed explanation of why they were chosen. Could the authors elaborate on the underlying motivation and principles for using these specific technologies? The current description in this area feels somewhat underdeveloped.

(2) Regarding "DIFFERENTIABLE DISCRETE SAMPLING," it seems that several of the components, such as the "Gumbel-Topk trick" and the "ECA module," are existing techniques. Could the authors clarify the novelty of their approach in this context and how these existing tricks are combined or adapted in a non-trivial way?

(3) The experiments primarily use CNN-based backbones. Have the authors evaluated or considered the applicability of their method to efficient Transformer architectures? The lack of such experiments may limit the perceived scope of the contribution.

---

> ### Author Response · Authors · 2025-11-24
> **Rebuttal to Reviewer 3mhu (Part 1)**
>
> Thank you for the time and effort you have dedicated to reviewing our manuscript. We are pleased to have the opportunity to address the three points you have raised.
>
> **Question 1: Insufficient Motivation for Technical Choices**
>
> We have elaborated on the motivation for introducing the Hadamard product in Chapter 1 and Section 2.2 of the manuscript. The primary motivation stems from a key issue in lightweight convolutional neural network design: addressing the computational redundancy in the expansion phase of the inverted bottleneck—a widely adopted paradigm for channel feature extraction, as demonstrated by GhostNet [1]. GhostNet mitigates this redundancy by generating new features through simple linear transformations, pioneering the concept of feature reuse and inspiring us to explore the integration of more efficient and concise operators into the inverted bottleneck computation.
>
> Furthermore, several other studies have applied the Hadamard product directly to the features themselves, deviating from the conventional gating mechanism approach. Among them, [2][3] explains from a kernel perspective how the Hadamard product enhances model expressiveness via implicit high-dimensional mapping. The lightweight and efficient nature of the Hadamard product motivated us to apply it to feature reuse.
>
> To enable the Hadamard product to function as a channel feature extractor, we propose the concept of "cross-Hadamard product," which involves evaluating importance scores to select and pair tensors from different channels for multiplicative interaction. This approach gradually forms the foundation of the entire ACH module.
>
> Regarding your question about the use of "HADAMARD FOR CHANNEL EXPANSION" and "DIFFERENTIABLE DISCRETE SAMPLING" in the Methodology section, we will address the rationale and necessity for these choices in the following response.
>
> **Question 2: Potentially Limited Novelty in Sub-components**
>
> The purpose of the ACH module is to achieve lighter and more effective cross-channel feature fusion through the cross-Hadamard product. Making the cross-Hadamard product learnable is the core contribution of this paper. The introduction of other modules is not aimed at incremental performance gains, but rather at enabling this learnability itself. In addition to the cross-Hadamard product, the entire ACH module incorporates four key components: PWConv, ECA, Gumble-TopK, and dySoft. Allow me to elaborate on the necessity of each:
>
> | Module      | Purpose                                                      |
> | ----------- | ------------------------------------------------------------ |
> | PWConv      | Follows the "feature reuse" design paradigm of GhostNet: establishes the most basic form of inter-channel feature interaction. |
> | ECA         | Continuous-to-Discrete: Converts continuous input tensors into channel weights used for making discrete selections. |
> | Gumble-TopK | Discrete-to-Continuous: Applies discrete selections to the continuous input tensors, using Gumbel noise to simulate continuity.<br/>*Bypasses Discreteness in Backpropagation:* Directly passes gradients to preceding modules (e.g., ECA), avoiding the non-differentiability of discrete operations. |
> | dySoft      | Stabilizes Training: Suppresses the multiplicative variance increase introduced by the cross-Hadamard product. |
>
> The ablation studies on Figure 3 in the manuscript validate the synergistic effect of this system, demonstrating that the absence of any single module compromises overall stability.

---

> ### Author Response · Authors · 2025-11-24
> **Rebuttal to Reviewer 3mhu (Part 2)**
>
> **Question 3: Limited Experimental Scope**
>
> The Multi-Head Self-Attention (MHSA) mechanism in Transformer focuses on the N-dimension, i.e., the relationships between tokens, while the Feed-Forward Network (FFN) operates on the C-dimension, integrating semantic information carried and aggregated within individual tokens. The FFN typically follows a classic inverted bottleneck structure, where the ACH module can effectively play a role in computational compression.
>
> To maintain research consistency—focusing on computer vision tasks and lightweight design—we have decided to supplement our experiments with improvements on the MobileViT model. Specifically, we replaced half of the FFN layers in all Transformer Encoders of MobileViT [4] with ACH modules. The comparative results are as follows:
>
> | Model                  | Params    | GFLOPs | Top1-Acc | Top5-Acc |
> | ---------------------- | --------- | ------ | -------- | -------- |
> | MobileViT-small        | 4,549,652 | 2.879  | 71.70    | 92.16    |
> | MobileViT-small-ACH    | 4,182,902 | 2.733  | 72.62    | 92.57    |
> | MobileViT-x-small      | 1,801,796 | 1.559  | 70.79    | 91.02    |
> | MobileViT-x-small-ACH  | 1,756,708 | 1.543  | 71.12    | 91.09    |
> | MobileViT-xx-small     | 878,164   | 0.588  | 70.07    | 90.89    |
> | MobileViT-xx-small-ACH | 860,172   | 0.582  | 70.67    | 90.92    |
>
> The experimental setup was configured as follows: the model was trained for 200 epochs with a batch size of 64, using CrossEntropyLoss as the criterion and AdamW as the optimizer with a momentum of 0.9 and weight decay of 1e-4. The learning rate was initialized at 0.003 and scheduled via a CosineAnnealingLR policy. All experiments were conducted on the CIFAR-100 dataset.
>
> The results demonstrate that replacing half of the FFN layers with the ACH module yielded significant improvements. Notably, this enhancement was achieved without increasing the number of parameters or computational complexity (FLOPs), leading to superior performance on the CIFAR-100 dataset compared to the baseline. This experiment substantiates that the ACH module exhibits a promising level of generalizability within deep learning, particularly for the role of a channel feature extractor.
>
> **Reference**
>
> [1] Kai Han, Yunhe Wang, Qi Tian, Jianyuan Guo, Chunjing Xu, and Chang Xu. Ghostnet: More
> features from cheap operations. In Proceedings of the IEEE/CVF conference on computer vision
> and pattern recognition, pp. 1580–1589, 2020.
>
> [2] Xu Ma, Xiyang Dai, Yue Bai, Yizhou Wang, and Yun Fu. Rewrite the stars. In Proceedings of the
> IEEE/CVF Conference on Computer Vision and Pattern Recognition, pp. 5694–5703, 2024.
>
> [3] Liangyu Chen, Xiaojie Chu, Xiangyu Zhang, and Jian Sun. Simple baselines for image restoration.
> In European conference on computer vision, pp. 17–33. Springer, 2022a.
>
> [4] Mehta, Sachin, and Mohammad Rastegari. "Mobilevit: light-weight, general-purpose, and mobile-friendly vision transformer." *arXiv preprint arXiv:2110.02178* (2021).

---

> > ### Comment · Reviewer_3mhu · 2025-11-27
> >
> > I would like to thank the authors for their response and clarifications. I have no further questions, and I will retain my initial positive rating.

---

### Official Review · Reviewer_ZaiK · 2025-10-27

**Soundness:** 2
**Presentation:** 3
**Contribution:** 3
**Rating:** 6
**Confidence:** 5

**Summary:**

This submission aims to achieve efficient and expressive feature recombination for lightweight vision models. It proposes Adaptive Cross-Hadamard (ACH) module, which leverages (i) learnable Hadamard (element-wise) products, (ii) discrete channel sampling via Gumbel-TopK, and (iii) dynamic softsign normalization for efficient feature expansion and recombination within CNNs. The optimal placement of ACH blocks alongside standard GhostNet-style linear expansion blocks is is discovered via differentiable neural architecture search to build the Hadaptive-Net.

Experiments are conducted on image classification (CIFAR-100, ImageNet-1K) and object detection (COCO) tasks. The results show Hadaptive-Net’s competitive accuracy-latency trade-offs compared to some baselines. Empirical analysis and theoretical complexity comparison such as ablation studies, architecture search, Grad-CAM visualizations are also conducted. Practically, it also presents engineering optimizations for GPU execution.

**Strengths:**

**(S1)** Clear drive and originality. This work identifies redundancy in existing lightweight vision models like GhostNet and inverted bottlenecks, which is a topic of practical signifance. It presents a mathematically grounded alternative in Sec. 3.1, leveraging the non-linear benefits of Hadamard products. This moves beyond previous linear cheap operations (e.g., GhostNet, gating-based models) and explores a new, parameter-free path to feature learning.

**(S2)** Technical soundness of method. The ACH module is well-engineered to address practical learning and stability constraints, including discrete channel selection via Gumbel-TopK sampling in Sec. 3.2, dynamic softsign normalization in Sec. 3.3, and NAS integration strategy in Sec. 3.4. In particular, I appreciate the ablation study in Fig. 3 which provides evidence for the design choices. The dramatic accuracy drop (73.57% to 64.39%) when replacing DySoft with vanilla BatchNorm is striking and supports the claims of stability. Appendix presents hyper-parameters, hardware, training schedules, and code release, which aids reproducibility.

**(S3)** Focus on deployment and GPU optimization. Section 4.2 shows efforts to bridge theoretical and real-world inference latency, such as custom CUDA kernels (direct-indexing and parity-balanced) and algorithmic choices for ACH operation. Fig. 5 visualizes runtime differences under varying tensor shapes. This focus on implementation details is useful for helping with real-world deployment in resource-constrained scenarios.

**(S4)** Broad experiments and competitive results. Ablation studies in Fig. 3 and Tab. 1, 2, 5 break down the contribution of each module component and validate plug-and-play integration on multiple efficient models. NAS-derived architecture choice is also examined in Tab. 3 and Appendix. Tab. 7 provides direct comparisons on CIFAR-100 and ImageNet-1K, considering both accuracy and efficiency metrics (latency, FLOPs). Notably, it achieves higher accuracy than recent MobileNetV4-L (74.73% vs. 74.38%) with less than 1/3 of FLOPs (669M vs. 2170M).

**(S5)** Visualization and interpretability. Grad-CAM in Fig, 6 and 7 show that ACH enables more semantically meaningful feature extraction and achieves improved early focusing compared to linear baselines. Diagrams like Fig. 2 (ACH pipeline), Tab. 1 (activation comparisons) are also well-constructed and informative.

**Weaknesses:**

**(W1)** In Appendix A.4, complexity ratio for Ghost/ACH modules is handled via “given $m \ll n$” simplifications, but experiment settings do not clarify the $m:n$ ratio in real deployments. This may hinder the reproduction or subsequent improvements in the community. I encourage the authors to provide more technical details on this point in revision.

**(W2)** For object detection in Tab. 6, the gains of Hadaptive-Net-L over GhostNetV3 or MobileNetV4 are a bit marginal. This raises the question whether it actually overcomes the efficiency wall or simply offers a modest incremental boost within a specific operator. I recommend the authors add direct clarification or discussion on this point in related section. This would significantly alleviate readers' concerns.

**(W3)** Seems no evidence is offered as to robustness under distribution/drift, input permutation, adversarial queries, or transfer to other domains (e.g., audio, language). This limits the “deep learning primitive” claim in Sec. 6. I recommend the authors consider slightly tempering this claim in the revised manuscript.

**(W4)** Incomplete literature review. Several important prior work [1] [2] [3] [4] [5] that applies Hadamard products in efficient vision models, attention/bilinear pooling, or edge detection are not included and discussed. I recommend the authors incorporate related discussions in the revised manuscript, which would not only improve the completeness but help emphasize the contributions and novelty of ACH within existing studies. It would be better to show the direct comparisons with [4] and [5].

**(W5)** The fine control of hyper-parameter $\tau$ is managed solely through heuristics and lacks sensitivity analysis experiments to support the choices.

---

## Reference

[1] Hadamard Product in Deep Learning: Introduction, Advances and Challenges, TPAMI 2025. This survey covers the use of Hadamard products in deep learning, especially for efficiency and nonlinear fusion.

[2] Hadamard Product for Low-rank Bilinear Pooling, ICLR 2017. It proposes efficient attention musing Hadamard products relevant to feature recombination methods.

[3] Unmixing Convolutional Features for Crisp Edge Detection, TPAMI 2021. It explores element-wise feature fusion similar to Hadamard approaches, directly related to efficient feature expansion.

[4] MogaNet: Multi-order Gated Aggregation Network, ICLR 2024. It is a representative efficient CNN model that uses gated Hadamard products for multi-order feature aggregation.

[5] Vision Mamba: Efficient Visual Representation Learning with Bidirectional State Space Model, ICML 2024. It first utilizes SiLU-gated Hadamard products within bidirectional state-space blocks to enable selective feature propagation.

**Questions:**

Most of my major concerns and related recommendations have been stated in the Weaknesses section. I encourage the authors to focus their efforts on addressing those points, as they are critical for strengthening the manuscript in the rebuttal stage.

The following are more specific, minor questions to help the authors think more deeply about certain design choices and experiment setups, which I hope might be helpful for this and future work:

**(Q1)** Could the authors provide theoretical analysis or insights into why dySoft normalization outperforms others in the ACH setting? Perhaps via variance/convergence analysis or more direct connections to gradient stability?

**(Q2)** Is there empirical evidence that ACH modules offer gains outside vision, such as language, audio? How do they perform in settings with heavy distribution shift? IMHO, such explorations may get closer to the heart of the question.

**(Q3)** It seems that the NAS leaves most early layers as Ghost modules and prefers ACH only at high dimensions. Do the authors see this as a limitation or a natural consequence of Hadamard product behavior? Is hybridization always preferable to pure ACH insertion? I believe this worths deeper investigation.


---

## Justification:

This submission presents strengths in motivation, technical soundness, thorough experiments, and practical engineering contributions. However, several concerns exist such as the lack of clarification for some technical details, limited literature review, and more. Overall, I am leaning towards acceptance and first give a rating of 6. I would be glad to raise my rating if thoughtful responses and improvements are provided. Conversely, if most of the concerns remain unaddressed, I may also lower my score. I am also open to follow-up discussions with the authors to help further strengthen this work.

I hope these comments help my fellow reviewers and ACs understand the basis of my recommendation.

---

> ### Author Response · Authors · 2025-11-24
> **Rebuttal to Reviewer ZaiK (Part 1)**
>
> Thank you for these insightful comments. Your suggestions are highly constructive and will significantly strengthen the rigor and clarity of this work. Below we provide a point-by-point response to the valuable issues you have raised.
>
> **(W1) Complexity Ratio Analysis**
>
> With regard to the computational complexity mentioned in Appendix A.4, we have carried out several groups of measured data on different input channel sizes and amplification ratios to confirm that ACH module has a very strong reasoning speed compared with the standard point-by-point convolution and ghost module.
>
> Point-wise convolution, Ghost module and ACH (Adaptive Cross-Hadamard) module are both channel feature extractors. They accept tensors with certain channel size, with giving amplified channel size tensors as output. The experiment is carried out for two specific situations: fixed 4-fold scaling ratio, 16-512 different input channel sizes; Fixed 64 input channel size, 4-24 times scaling ratio (224*224 per frame). We counted the Multiply-ACCumulate Operations (MACs) and Floating Point Operations (FLOPs) of the two groups of experiments as follows:
>
> ---
>
> https://i.ibb.co/PGkWz8Hs/channel-flops.png
>
> https://i.ibb.co/S4BrPrGC/channel-macs.png
>
> https://i.ibb.co/7JVrSZLY/ratio-flops.png
>
> https://i.ibb.co/rRBCh2Yg/ratio-macs.png
>
> ---
>
> The original analysis of the calculation efficiency ratio is indeed inaccurate. The formula:
>
> $$
> \text{Ratio}_\text{ACH}=\frac{m}{n}+\frac{1}{m}+\frac{1}{n}
> $$
>
> Simplification should consider $m<n,1\ll m,1\ll n$，then:
>
> $$
> \text{Ratio}_\text{ACH}=\frac{m}{n}
> $$
>
> This ratio can be shown intuitively in the above experimental results. From the experimental results, the computational complexity ratio under different channel sizes is relatively fixed, while different scaling ratios, which is $m/n$, show a linear relationship.
>
> **(W2) Object Detection Marginal Improvement**
>
> Regarding the extent of performance improvement on the object detection task, this point indeed helps clarify the primary contribution and positioning of our work.
>
> 1. **Core Objective and Rationale of the Experiment:** The primary objective of the object detection experiments in Table 6 was to validate the generalization capability of HadaptiveNet as a backbone network across different downstream tasks. As the reviewer is aware, the final performance of an object detector results from the synergistic interaction of the Backbone, Neck (feature fusion bottleneck), and Head. In this study, to purely and fairly evaluate the gain brought by the ACH module itself, we intentionally kept the Neck and Head identical to the baseline model. Under this configuration, HadaptiveNet achieved robust and consistent performance improvements under comparable parameter counts and computational complexity, which strongly validates its effectiveness as a backbone network. Differences in the design of components like the Neck are indeed a significant factor contributing to the performance gain being less pronounced than in the classification task.
> 2. **Clarification on "Breaking the Efficiency Bottleneck":** We posit that "breaking the efficiency bottleneck" manifests at two levels. This work focuses on the first level: within the existing paradigm of efficient network architectures, proposing a superior fundamental module (ACH) to construct a backbone network with stronger representational capacity under the same computational budget. This has been validated by its consistent improvements across multiple tasks, including classification and detection.
> 3. **Our Forward-Looking Perspective and Future Direction:** In fact, we fully concur with the deeper issue implied by the reviewer: maximizing the efficacy of the ACH module likely requires an end-to-end co-design. We have been aware of this and believe the more profound opportunity presented by this study lies in leveraging the principles of structured, lightweight cross-channel interaction embodied by ACH to redesign bottleneck components like Feature Pyramid Networks (FPNs), focusing on efficiently fusing multi-scale feature information. This represents a highly promising direction for breaking the efficiency bottleneck of current detectors. However, designing a novel, ACH-based efficient Neck is an equally important yet distinct and substantial research endeavor, requiring dedicated architectural exploration and extensive tuning. This paper focuses on the more fundamental innovation at the backbone level, aiming to provide the community with a powerful new foundation. We explicitly identify "designing an efficient Neck based on the ACH principles" as a clear and important future work, which will be elaborated upon in the discussion section of the paper to inspire subsequent research.

---

> ### Author Response · Authors · 2025-11-24
> **Rebuttal to Reviewer ZaiK (Part 2)**
>
> The current object detection results sufficiently demonstrate the superiority of the HadaptiveNet backbone. We appreciate the comment which will help readers more clearly understand the scope of our contribution. Concurrently, we will explicitly articulate our in-depth consideration of future directions in the paper – namely, how to extend the ACH innovation from the backbone to the entire detection system to fully realize its potential.
>
> **(W3) Lack of Evidence on Generality**
>
> You raise a valid point. The term "deep learning primitive" may overstate the current findings, as our empirical validation is thus far confined to downstream tasks in computer vision. Nevertheless, we have conducted additional experiments to preliminarily explore the potential of the ACH module in the NLP domain with following experiments. For details, please refer to our response to Question 2.
>
> **(W4) Related Work**
>
> We have carefully reviewed the recommended literature, which indeed constitutes important related work.
>
> Work [1] proposes the first taxonomy for the application of the Hadamard product in deep learning, categorizing its uses into four classes: high-order interactions, multimodal fusion, adaptive modulation, and efficient operators. StarNet, which served as a motivational example in our writing, is an application of the Hadamard product in efficient operators and high-order interactions. Work [2] represents a classic example of Hadamard product application in multimodal fusion. It addresses feature fusion in multimodal learning—particularly in Visual Question Answering (VQA)—by employing the Hadamard product to achieve low-rank bilinear pooling as an approximation of full bilinear pooling. Common gating mechanisms are typical applications in adaptive modulation; for instance, work [3] refines pixel-level feature fusion by applying the Hadamard product between an edge heatmap and a context weight map learned via an attention mechanism. The ACH module proposed in our work, motivated by replacing redundant computations in inverted bottlenecks, follows the research direction of GhostNet as an application in efficient operators. In terms of its behavioral characteristics, it also shows strong potential for applications in multimodal fusion, such as constructing bottlenecks for object detectors or improving low-rank bilinear pooling processes.
>
> MogaNet [4] offers a clear perspective on modern convolutional neural networks, categorizing operations into spatial feature extractors—primarily convolution and TRM—and channel feature extractors, led by MLP-based inverted bottlenecks. However, MogaNet's use of the Hadamard product remains oriented toward static self-interactive gating and scaling. For example, expressions of the form $\gamma_s\odot(Y-GAP(Y))$ belong to scaling-controlled aggregation ratios, while forms like $\text{SiLU}(\text{Conv}(X))\odot\text{SiLU}(\text{Conv}(Y_C))$ involve self-interactive gating, where $Y_C$ is derived only from corresponding channels of $X$ via depthwise separable convolution and other independent transformations, without direct cross-channel interaction. The key focus of our work—leveraging the Hadamard product to improve channel context extraction and making the combinatorial selection operation learnable—remains outside its scope. That said, it is noteworthy that MogaNet's channel feature extractor captures inter-channel relations through direct linear projections, which may inspire a more streamlined design for the evaluation network within our ACH module.
>
> Manba represents a novel spatial feature extractor alongside convolution and MHSA. Unlike the linear time-invariant nature of convolution and MHSA, Manba dynamically generates key parameters based on the input tensor. Vision Manba [5] adapts the Manba block to computer vision following the Vision Transformer (ViT) approach. The use of the Hadamard product in Manba and Vision Manba appears in its core recurrence formulas: $h_t=\overline A_t\odot h_{t-1}+\overline B_t\odot x_t$ and $y_t=C_t\odot h_t$, where $\overline A_t,\overline B_t\in\mathbb R^{N\times S}$ are learnable vectors determined along the sequence dimension, and $h_t\in\mathbb R^{N\times S}$ is a dynamically computed vector along the sequence dimension. It can be said that Manba's dynamic Hadamard product underlies a spatial feature extractor based on a linear time-varying mechanism, whereas our ACH's dynamic Hadamard product underlies a channel feature extractor based on a discrete selection mechanism.
>
> We sincerely appreciate you bringing these papers to our attention. They are highly valuable for contextualizing our contributions, and we will incorporate discussions of them into the related work section of the manuscript.

---

> ### Author Response · Authors · 2025-11-24
> **Rebuttal to Reviewer ZaiK (Part 3)**
>
> **(W5) Hyperparameters Choice**
>
> Since the ACH module in this paper draws on the training approach of GDAS (Gradient-based search using Differentiable Architecture Sampler), the training process is often subject to second-order control, which implies that there are numerous hyperparameter options during the temperature modulation process of Gumbel-TopK. We acknowledge that further hyperparameter analysis is required here.
>
> Reviewing the second-order modulation process here:
>
> ```py
> def _adjust_tau_with_grad(self, grad: Tensor):
>     if self.tau_adj.data != 0 and grad is not None:
>         grad_norm = grad.norm()
>         alpha = self.tau_alpha
>         alpha *= self.tau_relax if self.tau_adj <= grad_norm else -self.tau_tight
>         self.tau.data = torch.clamp(self.tau * (1.0 + alpha), max=self.tau_clamp_max, min=self.tau_clamp_min)
>     if grad is not None:
>         self.tau_adj.data = grad.norm()
> ```
>
> It can be observed that the adjustable hyperparameters include `relax`, `tight`, `alpha`, `clamp_max`, and `clamp_min`. Recalling the debugging data from our previous exploration of optimal hyperparameters, as shown in the table below:
>
> | relax | tight | alpha  | clamp_max | clamp_min | CIFAR-100 Acc. |
> | ----- | ----- | ------ | --------- | --------- | -------------- |
> | 1.0   | 1.0   | 0.005  | 4.0       | 0.01      | **73.96%**     |
> | -     | -     | -      | -         | 0.001     | 73.72%         |
> | -     | -     | 0.0005 | -         | -         | 73.47%         |
> | -     | -     | 0.002  | -         | -         | 73.48%         |
> | -     | -     | 0.02   | -         | -         | 73.66%         |
> | 0.95  | -     | -      | -         | -         | 73.76%         |
>
> It can be observed that the current hyperparameter combination is relatively superior to others. However, there remains considerable scope for further exploration.
>
> ---
>
> **(Q1) DySoft Indispensability**
>
> To be honest, I cannot explain this problem with quantitative experiments, because the model without dySoft training is extremely unstable and has no representative experimental data. However, these phenomena can illustrate a problem that dySoft is empirically necessary. Next, I will explain this problem through the convergence analysis of statistical representation.
>
> **Problem Analysis**
>
> Traditional normalization methods, such as BatchNorm and LayerNorm, have a priori assumption that the statistical mean and statistical variance of the tensors they receive are knowable and traceable, which constitutes the basis of model convergence. In the process of ACH training and reasoning, we will involve a standard $\mathrm {Z}_i \odot\mathrm {Z}_j $ cross Hadamard product calculation. In previous machine learning methods, the use of Hadamard product is usually self referential, that is, $Z^2=\mathrm{Z}\odot\mathrm{Z}$. In this case, we can easily infer the mean value of $Z^2=\mathrm{Z}\odot\mathrm{Z}$ from the mean and variance $\mu,\sigma^2$ of $Z$ :
>
> $$
> \begin{array}{l}
> \text{Var}(Z)=\text E[(Z-\mu)^2]=\text E[Z^2]-(\text E[Z])^2\\
> E[Z^2]=\mu^2+\sigma^2
> \end{array}
> $$
>
> Since tensor $Z$ was processed by normalization from above layer, which approximately satisfies $Z\sim N(\mu,\sigma^2)$. According to the fourth moment formula of normal distribution:
>
> $$
> \text E[Z^4]=\mu^4+6\mu^2\sigma^2+3\sigma^4
> $$
>
> $$
> \text{Var}(Z^2)=\text E[Z^4]-(\text E[Z^2])^2=(\mu^4+6\mu^2\sigma^2+3\sigma^4)-(\mu^2+\sigma^2)^2=2\sigma^2(2\mu^2+\sigma^2)
> $$
>
> If the self referring Hadamard product is deformed, for example $\phi_1(Z)\odot \phi_2(Z)$，Let $\phi$ here be a linear transformation operator, the corresponding matrix form is $X_1,X_2\ (X\in\mathbb R^{m\times n})$, bias vectors are $b_1,b_2\ (b\in\mathbb R^{m})$, then:
>
> $$
> \text E[\phi(Z)]=\text E[XZ+b]=\mu\cdot\dfrac{\sum_i^m\sum_j^nX_{i,j}}{m}+\text E[b]
> $$
>
> For variance, since $Z$ can approximate normal distribution, here we assume that its elements are i.i.d, then there are:
>
> $$
> \text{Var}(\phi(Z))=\dfrac{1}{m}\cdot\sum_i^m\text{Var}(\phi(Z)_i)=\dfrac{1}{m}\cdot \sum_i^m \sum_j^n A^2\cdot\sigma^2=\sigma^2\cdot\dfrac{\|A\|^2_F}{m}
> $$

---

> ### Author Response · Authors · 2025-11-24
> **Rebuttal to Reviewer ZaiK (Part 4)**
>
> Suppose $\phi$ is a nonlinear transformation operator, which does not directly exist the predictability of analytical solutions. However, the purpose of normalization method is not to accurately track the statistical representation of tensors, but to ensure that the statistical representation of tensors remains stable in the reasoning process.
>
> Let the mapping $T_f:\mathbb R\times\mathbb R_{>0}\to\mathbb R\times\mathbb R_{\ge0}$ be: $(\mu, \sigma^2)\mapsto(\mu',\sigma'^2)$. If $T_f$ unbounded, that is, there is a sequence $(\mu_k,\sigma_k^2)$ such that $\|T_f(\mu_k,\sigma_k^2)\|\to\infty$ as long as a layer accidentally reaches the state (such as disturbance, initialization deviation), the next layer of statistics will be unstable; If $T_f$ is discontinuous or the derivative is unbounded (e.g. $f(z)=\mathrm 1_{z>0}$ is at $\mu=0$), small disturbance can lead to $\mu',\sigma'^2$ upheaval, resulting in unstable training.
>
> BatchNorm is generally considered in CV tasks. BN independently estimates the mean and variance of $k$ for each channel:
>
> $$
> \hat\mu_k=\mathbb E_{\mathrm x\sim\mathcal B}[x_k],\quad\hat\sigma^2_k=\text{Var}_{\mathrm x\sim\mathcal B}(x_k)
> $$
>
> And perform channel by channel affine transformation:
>
> $$
> x_k'=\gamma_k\cdot\dfrac{x_k-\hat\mu_k}{\sqrt{\sigma^2_k+\epsilon}}+\beta_k
> $$
>
> This operation does not force statistical consistency between channels, but allows or even encourages significant statistical heterogeneity between channels:
>
> $$
> \exists\ i\neq j\quad\text{s.t.}\quad\hat\mu_i\neq\hat\mu_j,\ \hat\sigma^2_i\neq\hat\sigma^2_j
> $$
>
> This property is consistent with the inductive bias of "channel division" in convolutional networks - different channels can professionally respond to different semantic patterns (such as edge, texture, color), which is the key basis for its high representation efficiency in visual tasks. In contrast, LN is normalized in the sample dimension:
>
> $$
> \mathrm{x'}=\gamma\cdot\dfrac{\mathrm x-\mu}{\sigma}+\beta,\quad\mu=\dfrac{1}{C}\sum_k^Cx_k,\ \sigma^2=\dfrac{1}{C}\sum_k^C(x_k-\mu)^2
> $$
>
> The implicit priori is that all channels at the same spatial location should have the same statistical scale, which drives statistical convergence between channels. This assumption is compatible with the inductive bias of "all tokens are comparable" in the global attention mechanism (such as ViT), but in CNN dominated by local receptive fields, it will weaken the channel specific characterization ability and lead to performance degradation.
>
> Let us consider $y_{ij}=x_i\odot x_j$, its output statistics depend on the joint second moment of the input channel. Under the heterogeneity distribution induced by BN, let  $x_i\sim\mathcal N(\mu_i,\sigma^2_i),\ x_j\sim\mathcal N(\mu_j,\sigma^2_j)$ and i.i.d., then:
>
> $$
> \mathbb{E}[y_{ij}]=\mu_i\mu_j\\
> \text{Var}(y_{ij})=\mu^2_i\sigma^2_j+\mu^2_j\sigma^2_i+\sigma^2_i\sigma^2_j
> $$
>
> When the channel statistics differ significantly (e.g. $|\mu_i|\gg|\mu_j|$ or $\sigma_i\gg\sigma_j$), the variance shows a multiplicative amplification effect, which is far beyond the single channel scale range. The affine parameters of BN are only channel specific, which can not effectively correct the new statistical offset caused by such cross-channel coupling. Otherwise, the pairing process of $i, j$ is obtained by the nonlinear transformation of each input, which makes it impossible for the statistical representation iterative map $T_f$ to find the fixed point.
>
> Although LN normalization may be used inside ACH module, it is very important to understand the heterogeneity between channels in CV tasks. There is usually a typical CNN trunk containing BN upstream of the module, so the whole feature learning process has been dominated by the heterogeneity of BN a priori. The model's understanding of image semantics will evolve spontaneously towards the direction of "channel specialization". At this time, if a strong cross-channel nonlinear module with implicit homogeneity assumption is inserted into the reasoning chain, it will lead to a priori conflict.
>
> **Solution**
>
> The dySoft we introduced is essentially a variant of the softsign activation function:
>
> $$
> y = \cfrac{\alpha x}{1+|\alpha x|}\cdot w+b,\quad\lim_{\alpha x\to\pm\infty}\cfrac{\alpha x}{1+|\alpha x|}=\pm1
> $$
>
> Due to the boundedness of softsign, no matter how large the input variance $\sigma^2$ is, the output variance is rigidly limited in the $(0,1)$ range; When the input is small, it shows approximate linearity and maintains the characteristics of the signal. The parameter $\alpha$ can dynamically balance the expression and compression of the layer.

---

> ### Author Response · Authors · 2025-11-24
> **Rebuttal to Reviewer ZaiK (Part 5)**
>
> When the cross-Hadamard $y_{ij}=x_i\odot x_j$ has variance like $\sigma_i,\sigma_j\gg1$, $\text{Var}(y_{ij})$ increased by $\mathcal O(\sigma^4)$. After accessing dySoft, this trend can be significantly compressed and given boundedness. In addition, dySoft is also designed based on the hypothesis of channel heterogeneity, which is a priori compatible with the heterogeneity of BN. Its $w, b$ parameters are channel specific, and can independently learn the scale and offset for each cross-Hadamard product channel. At the same time, it does not destroy the channel professional representation established by the upstream BN, and only makes local intervention on the "danger signal", thus realizing the organic unity of characterization and stability.
>
> In summary, dySoft is a learnable statistical compression gating (SCG) module, which achieves hard variance clamping for high square error input through bounded nonlinear mapping $\mathcal S(u)=u/(1+|u|)$ and restores the characterization capacity in combination with channel specific affine transformation. Without violating the heterogeneity prior of batch normalization, the design effectively inhibits the growth of multiplicative variance caused by cross channel nonlinearity (such as cross Hadamard product), and makes the statistical map $T：(\mu, \sigma^2)\mapsto(\mu',\sigma'^2)$ bounded and smooth, so as to meet the core condition of "knowability", providing a stable and convergent statistical target for the normalization layer.
>
> **(Q2) Prof of Generality**
>
> We tried to extend ACH module to NLP and did the following experiments:
>
> | Layers  | Modify              | In-Out-Channel | Mid-Channel | SST-2 | MNLI-10% |
> | ------- | ------------------- | -------------- | ----------- | ----- | -------- |
> | BERT-6  | FFN (fixed channel) | 764            | 764         | 64.7  | 36.7     |
> | BERT-6  | ACH Module          | 764            | 2043        | 76.2  | 41.8     |
> | BERT-6  | FFN                 | 764            | 2048        | 80.3  | 46.7     |
> | BERT-12 | FFN (fixed channel) | 764            | 764         | 53.1  | 34.3     |
> | BERT-12 | ACH Module          | 764            | 2043        | 75.9  | 41.2     |
> | BERT-12 | FFN                 | 764            | 2048        | 80.6  | 45.5     |
>
> We conducted a comparative test on the 6-layer and 12 layer BERT [6]. The accuracy of the BERT model using different channel feature extractors (FFN, ACH module and standard FFN with unchanged middle layer dimension) was tested in SST-2 and 10% MNLI datasets.
>
> The models were evaluated on two standard natural language understanding benchmarks: the Stanford Sentiment Treebank (SST-2) [7] for binary sentiment classification and the Multi-Genre Natural Language Inference (MNLI) [8] dataset for textual entailment. For SST-2, the model was trained and evaluated on the full dataset. To assess performance in a low-resource setting, the model was trained on a 10% stratified subset of the MNLI training set and evaluated on the full matched validation set.
>
> The training configuration was consistent across both tasks. Models were trained for 3 epochs with a global batch size of 32 and evaluated with a batch size of 64. The optimization used a learning rate of 2e-5 with a linear warmup over the first 10% of the training steps and weight decay of 0.01. The models, which followed a BERT-base architecture (12 layers, 12 attention heads, 768-dimensional hidden states), were initialized with random weights. Input sequences were tokenized using the `bert-base-uncased` tokenizer with a maximum length of 128 tokens. The sole evaluation metric was classification accuracy, calculated as the percentage of correctly predicted labels against the ground truth. All experiments were run on a single GPU without mixed-precision training.
>
> As a result, the performance of ACH module was not stunning enough to exceed the BERT baseline. However, from the perspective of the motivation of compressing the calculation scale, ACH module still plays a big role. Only adding a small amount of cross-Hadamard product in one step can approach the FFN without channel depth to a better level, revealing its potential as a unique algorithm in the field of NLP.
>
> As we are mainly engaged in CV related work and lack relevant experience in NLP field, the experimental setup may be a little rough. If there are any problems, you are welcome to correct them.

---

> ### Author Response · Authors · 2025-11-24
> **Rebuttal to Reviewer ZaiK (Part 6)**
>
> **(Q3) Dimension Preference**
>
> Our NAS experiments (Table 3) and layer replacement studies (Table 2) demonstrate that NAS tends to favor using ACH modules in the high-dimensional stages of the network while retaining pointwise convolutions in the low-dimensional stages. The results indicate that this hybrid design outperforms a pure ACH-based architecture, which we argue is a natural consequence of traditional CNN design principles.
>
> CNN architectures follow a consistent paradigm when processing low- and high-dimensional features. Convolution excels at capturing spatial patterns and texture details. In low-dimensional layers, where feature information is concentrated within individual images and channel count is limited, convolution operates effectively. In contrast, as feature dimensionality increases, information becomes distributed across channels. Here, the pointwise convolution within inverted bottlenecks—essentially a feed-forward network—plays the dominant role by integrating cross-channel information. A similar rationale underlies Transformer design: MHSA focuses on token-token relationships, while the FFN integrates semantic information within and across tokens.
>
> The FFN without activation functions acts as a linear transformation layer, or more precisely, it introduces non-linearity only within high-dimensional spaces statically, rather than during the process of channel feature interaction. The ACH module addresses this issue by introducing cross-channel Hadamard products, enabling more refined feature interaction decisions. However, one design motivation of ACH is to reduce computational redundancy during channel expansion. Its feature reuse mechanism inherently prioritizes essential interactions at the cost of neglecting some secondary channels. When processing low-dimensional tensors with fewer channels, features tend to exhibit stronger linear independence (lower redundancy). ACH’s computational approach, which assumes a certain degree of redundancy a priori, may thus struggle to capture subtle inter-channel variations in early-stage layers.
>
> To address this limitation, we intend to explore in future work how new architectures can better balance redundancy and essential feature ratios in early layers, thereby broadening the applicability of cross-channel Hadamard products.
>
> Finally, thank you again for your patience in reviewing our work!
>
> **Reference**
>
> [1] Hadamard Product in Deep Learning: Introduction, Advances and Challenges, TPAMI 2025.
>
> [2] Hadamard Product for Low-rank Bilinear Pooling, ICLR 2017.
>
> [3] Unmixing Convolutional Features for Crisp Edge Detection, TPAMI 2021.
>
> [4] MogaNet: Multi-order Gated Aggregation Network, ICLR 2024.
>
> [5] Vision Mamba: Efficient Visual Representation Learning with Bidirectional State Space Model, ICML 2024.
>
> [6] Devlin, Jacob, et al. "Bert: Pre-training of deep bidirectional transformers for language understanding." *Proceedings of the 2019 conference of the North American chapter of the association for computational linguistics: human language technologies, volume 1 (long and short papers)*. 2019.
>
> [7] Socher, Richard, et al. "Recursive Deep Models for Semantic Compositionality Over a Sentiment Treebank." *Proceedings of the 2013 Conference on Empirical Methods in Natural Language Processing*, 2013, pp. 1631-42.
>
> [8] Williams, Adina, et al. "A Broad-Coverage Challenge Corpus for Sentence Understanding through Inference." *Proceedings of the 2018 Conference of the North American Chapter of the Association for Computational Linguistics: Human Language Technologies, Volume 1 (Long Papers)*, 2018, pp. 1112-22.

---

> > ### Comment · Reviewer_ZaiK · 2025-11-26
> > **Official Response by Reviewer ZaiK**
> >
> > Dear Authors,
> >
> > I respect the great efforts the authors have put into this rebuttal. It is not common to see such a precise response that includes not only detailed clarifications but extra experiments on different modalities and hardware profiling within the short rebuttal window. This level of engagement significantly strengthens the submission.
> >
> > The rebuttal addresses most of my concerns. Below are my comments to several important points to ensure these valuable insights are can be effectively incorporated into final version of the paper.
> >
> >
> > **(Q1) DySoft's justification.**
> > The explanation about statistical heterogeneity of BN vs. homogeneity assumption of LN, and how DySoft bridges this gap for Hadamard products is excellent. In particular, the argument that standard Hadamard products cause multiplicative variance explosion, which BN cannot correct due to affine limitations provides the reason for your method. Please ensure this derivation and the intuition about the boundedness via softsign will be added to the manuscript.
> >
> >
> > **(W3, Q2) Generalization and NLP experiments.**
> > I am impressed that you managed to run BERT experiments on SST-2 and MNLI. The efficiency savings (e.g., approaching FFN performance with significantly fewer parameters) well support the claim that ACH is a viable "primitive" for channel interaction beyond vision. I recommend adding these results to the Appendix with the title like "Preliminary Exploration in NLP."
> >
> >
> > **(W2) Object Detection and Future Work:**
> > Your response regarding the "efficiency bottleneck" at the Neck/Head level is well-reasoned.
> > Suggestion: Please explicitly include the discussion about the need for co-designing the Neck (FPN) with ACH-based principles in the "Discussion" or "Future Work" section. This clarifies for future readers why the mAP gains might not scale linearly with the backbone improvements and sets a clear roadmap for the community.
> >
> >
> > **(W1) Complexity analysis.**
> > From my perspective, the additional MACs/FLOPs plots links is much more convincing than the original one. Please replace or augment the simplified complexity formula in Appendix with the new ones. Real-world latency and MACs are what practitioners care about most.
> >
> > Overall, the difference between this work and existing methods is now clear. Given the technical soundness, the robustness added by the new experiments, and the clarifications in the rebuttal, I increase my rating to 8. I hope all these clarifications, discussions, and new results can be incorporated in to the next version of manuscript. IMHO, these details would largely help readers better understand this paper and may lead to further thoughts and insights. I look forward to the next version of this work.
> >
> >
> >
> > Best regards,
> >
> > Reviewer ZaiK

---

> ### Comment · Reviewer_ZaiK · 2025-11-26
> **Official Response by Reviewer ZaiK (Part 2)**
>
> In addition, I recommend writing a global response after completing all the updates, listing all significant clarifications, modifications, and updates one by one. Also, it would be helpful to explicitly note if you think their concerns have been addressed in rebuttal (even if the reviewer has not yet replied), and would also help the ACs understand the current status, your improvements, and the post-rebuttal quality of this work.

---

> > ### Author Response · Authors · 2025-11-28
> >
> > Dear Reviewer ZaiK,
> >
> > Thank you very much for your recognition of our work and your valuable suggestions! We will incorporate all the important clarifications, additional experiments, and related insights mentioned in your comments into the main text and appendix of the paper in an appropriate form. At the same time, we will summarize the responses to all comments during this rebuttal phase, along with the revisions and updates made to the paper, and provide a unified explanation in the global comment section. The latest revised version of the manuscript will also be submitted for review.
> >
> > Once again, we extend our sincere gratitude!
> >
> > Sincerely,
> >
> > The Authors

---

### Official Review · Reviewer_qj3w · 2025-10-27

**Soundness:** 4
**Presentation:** 3
**Contribution:** 3
**Rating:** 6
**Confidence:** 4

**Summary:**

The authors introduce the Adaptive Cross-Hadamard (ACH) module, utilizing channel attention-guided feature gating and differentiable discrete sampling along with Hadamard (element wise). This module, when stacked can efficiently and effectively model nonlinear representations and high-dimensional relationships. The authors develop a novel and effective training methodology utilizing the Gumbel TopK trick to limit the compute of the model across channels as well as a novel hardware-efficient normalization scheme due to the nature and requirements of this new model. The authors use architecture search to demonstrate that this module is preferred over existing methods and demonstrate its computational efficiency. Finally, especially at small scale, the authors Hadaptive-Net models clearly outperform competing, less efficient models on top1 accuracy for the CIFAR-100 and ImageNet1K datasets. As a result, the authors have obtained a more efficient, more effective model.

**Strengths:**

Through the introduction of the Adaptive Cross-Hadamard (ACH) module these results develop a novel method for training pairwise Hadamard products with very few parameters and utilize Hadamard operations as building blocks for efficient vision models, which is relatively unexplored. Moreover the authors' module offers parameter-free feature reuse, reducing computational cost. The ACH module has strong accuracy/speed trade-offs, appealing for real-world and edge deployment and a hardware friendly implementation. Finally, in evaluation the ACH based classification models offer state of the art accuracy/speed tradeoffs.

**Weaknesses:**

I think it would be very helpful to have a precise description of the end to end method listed as an algorithm at some point. In particular, equation 10 is somewhat ambiguous to me. It is not very clear how the tensor X is added to the cross Hadamard product (e.g. what is the $\oplus$ operator doing?) It is also  somewhat confusing at first blush what components of the model are trainable and which aren't. Finally the dysoft operator is somewhat ambiguously applied to the output of the pairwise Hadamard computation.

**Questions:**

Please provide a clear and concise mathematical description of the algorithm, preferably with descriptions of the input and output dimensions of the individual operators. More precise definitions and clear descriptions of the algorithm will make this paper much stronger.

 Additionally, do the authors believe this method would be useful for tasks beyond classification? If so, please provide some justification or evidence.

---

> ### Author Response · Authors · 2025-11-24
> **Rebuttal to Reviewer qj3w (Part 1)**
>
> Thank you so much for your review comments! We will provide answers based on the questions you have raised. Firstly, regarding the mathematical description of the algorithm, we acknowledge that it is indeed somewhat complex, but combining it with code implementation may help you better understand our algorithm.
>
> To clarify the algorithmic process and address the concern regarding the input/output dimensions of individual operators, we provide a step-by-step breakdown of the `AdaptiveCrossHadamard` module. We map the code implementation directly to the tensor transformations.
>
> Let the input tensor be $\mathbf{X}$ with shape $(B, C, H, W)$, where $B$ is batch size, $C$ is the input channel dimension, and $H, W$ are spatial dimensions. The target number of selected channels is $C_{(s)}$.
>
> **Step 1: Pre-processing and Normalization**
> The input features first pass through a $1\times1$ convolution and Batch Normalization. This retains the original dimensions but adapts the features for selection.
>
> * **Code Reference:**
>
>   ```python
>   # self.fc = nn.Conv2d(c1, c1, 1)
>   x = self.fc(x)
>   x = self.norm_x(x)
>   ```
>
> * **Tensor Transformation:**
>   Input: $(B, C, H, W) \to$ Output: $(B, C, H, W)$
>
> **Step 2: Channel Importance Scoring**
> An Efficient Channel Attention (ECA) module (`eva_net`) computes a score for each channel. The output is flattened to produce a vector of logits representing channel importance.
>
> * **Code Reference:**
>
>   ```python
>   # self.eva_net = ECA(5, True)
>   logits = self.eva_net(x).flatten(1)
>   ```
>
> * **Tensor Transformation:**
>   Input: $(B, C, H, W) \to$ Output: $(B, C)$
>
> **Step 3: Adaptive Channel Selection**
> Based on the logits, we select the top-$C_{(s)}$ most salient channels.
>
> * **Training Phase:** We use Gumbel-Softmax to generate a differentiable mask (`mask_mat`) and perform matrix multiplication to extract features.
>
> * **Inference Phase:** We use discrete indexing (`topk`) for efficiency.
>
> * **Code Reference:**
>
>   ```python
>   # Training: Differentiable extraction
>   mask_mat = ... # Generated via Gumbel-Softmax
>   x_sel = (mask_mat @ x.flatten(2)).view(_shape)
>
>   # Inference: Discrete indexing
>   _, topk_idx = torch.topk(logits, self.cs)
>   x_sel = x[batch_idx, topk_idx]
>   ```
>
> * **Tensor Transformation:**
>   Input: $(B, C, H, W)$ and Logits $(B, C) \to$ Output $\mathbf{X}_{sel}$: $(B, C_{(s)}, H, W)$
>   *(Note: The channel dimension is reduced from $C$ to $C_{(s)}$)*
>
> **Step 4: Cross-Hadamard Expansion**
> This is the core operation. We perform element-wise multiplication (Hadamard product) between all unique pairs of the selected $C_{(s)}$ channels. The index pairs $(i, j)$ are pre-computed in the `__init__` method.
>
> * **Code Reference:**
>
>   ```python
>   # self.hadamard_i and self.hadamard_j are pre-calculated index buffers
>   x_sel_ex = x_sel[:, self.hadamard_i, ...] * x_sel[:, self.hadamard_j, ...]
>   ```
>
> * **Tensor Transformation:**
>
>   Input $\mathbf X_{sel}$: $(B, C_{(s)}, H, W)$
>
>   The number of expanded channels is calculated as $C_{exp} = \frac{C_{(s)} \times (C_{(s)} - 1)}{2}$.
>
>   Output $\mathbf X_{ex}$: $(B, C_{exp}, H, W)$
>
> **Step 5: Fusion and Output**
> The expanded second-order features are normalized and concatenated with the original first-order features.
>
> * **Code Reference:**
>
>   ```python
>   x_sel_ex = self.norm(x_sel_ex)
>   return torch.cat([x, x_sel_ex], dim=1)
>   ```
>
> * **Tensor Transformation:**
>   Input 1: $(B, C, H, W)$
>   Input 2: $(B, C_{exp}, H, W)$
>   **Final Output:** $(B, C + C_{exp}, H, W)$
>
> **Summary Table of Dimensions**
>
> | Operator        | Input Dimension                      | Output Dimension         | Description                              |
> | :-------------- | :----------------------------------- | :----------------------- | :--------------------------------------- |
> | `fc` & `norm_x` | $(B, C, H, W)$                       | $(B, C, H, W)$           | Linear projection & Norm                 |
> | `eva_net`       | $(B, C, H, W)$                       | $(B, C)$                 | Importance Scoring                       |
> | `_get_selected` | $(B, C, H, W)$                       | $(B, C_{(s)}, H, W)$     | Dimensionality Reduction ($C_{(s)} < C$) |
> | Cross-Hadamard  | $(B, C_{(s)}, H, W)$                 | $(B, C_{exp}, H, W)$     | Feature Expansion                        |
> | `torch.cat`     | $(B, C, \dots), (B, C_{exp}, \dots)$ | $(B, C + C_{exp}, H, W)$ | Feature Fusion                           |

---

> > ### Comment · Reviewer_qj3w · 2025-11-25
> >
> > Thanks for this explanation, I have no other questions.

---

> ### Author Response · Authors · 2025-11-24
> **Rebuttal to Reviewer qj3w (Part 2)**
>
> Review the formula 10 you mentioned:
> $$
> \mathbf{Y}=\mathbf{X}\oplus\\{\mathbf Z_i\odot\mathbf Z_j\mid\\{(i,j)\in\{1,2,\dots,C_{(\text{s})}\}, i\neq j\\}\\}\\
> \text{s.t.}\quad\mathbf{Z}=\mathbb{M}\cdot\mathbf{X}
> $$
> is a summary of the whole calculation process. $\mathrm Z$ or $\mathrm X_{sel}$ are $C_{(s)}$ important tensors filtered from the input tensor $\mathrm X$ along the channel dimension $C$. And $\mathbb M\in\mathbb R^{B\times C_{(s)}\times C}$ is the selection matrix which carrying the gradient, corresponding to variant `mask_mat` in the above code.
>
> Regarding the second question on applicability beyond classification, we confirm our method's potential in other domains. Section 5.4 of our paper demonstrates Hadaptive-Net's effectiveness in object detection on COCO dataset (Table 6), where our approach improved mAP\@0.5:0.95 by 0.5 over MobileNetV3-S with lower computational cost.
>
> Furthermore, we conducted preliminary experiments extending the ACH module to NLP tasks by replacing feed-forward networks (FFN) in BERT [1] architectures. As shown in the table below, when tested on SST-2 [2] sentiment analysis and MNLI [3] textual entailment tasks:
>
> | Model   | Feature Extractor   | In-Out-Channel | Mid-Channel | SST-2 Acc. | MNLI-10% Acc. |
> | ------- | ------------------- | -------------- | ----------- | ---------- | ------------- |
> | BERT-6  | FFN (fixed channel) | 764            | 764         | 64.7       | 36.7          |
> | BERT-6  | ACH Module          | 764            | 2043        | 76.2       | 41.8          |
> | BERT-6  | FFN                 | 764            | 2048        | 80.3       | 46.7          |
> | BERT-12 | FFN (fixed channel) | 764            | 764         | 53.1       | 34.3          |
> | BERT-12 | ACH Module          | 764            | 2043        | 75.9       | 41.2          |
> | BERT-12 | FFN                 | 764            | 2048        | 80.6       | 45.5          |
>
> While the ACH module doesn't yet match the full performance of standard expanded FFNs, it achieves substantial improvements over fixed-channel networks with significantly fewer parameters. Specifically, the ACH module approaches 95% of the performance of standard FFNs while using approximately 40% fewer parameters. This demonstrates its potential as an efficient feature expansion operator across modalities. We acknowledge our expertise is primarily in vision tasks and welcome collaboration with NLP specialists to further refine these applications.
>
> **Reference**
>
> [1] Devlin, Jacob, et al. "Bert: Pre-training of deep bidirectional transformers for language understanding." *Proceedings of the 2019 conference of the North American chapter of the association for computational linguistics: human language technologies, volume 1 (long and short papers)*. 2019.
>
> [2] Socher, Richard, et al. "Recursive Deep Models for Semantic Compositionality Over a Sentiment Treebank." *Proceedings of the 2013 Conference on Empirical Methods in Natural Language Processing*, 2013, pp. 1631-42.
>
> [3] Williams, Adina, et al. "A Broad-Coverage Challenge Corpus for Sentence Understanding through Inference." *Proceedings of the 2018 Conference of the North American Chapter of the Association for Computational Linguistics: Human Language Technologies, Volume 1 (Long Papers)*, 2018, pp. 1112-22.

---

### Author Response · Authors · 2025-12-01
**Summary of Rebuttal Period**

During the rebuttal period, we thoroughly addressed the reviewers' concerns and have now reached a consensus with all reviewers. We provide below a summary of our responses, organized into four key themes: Coherence, Theoretics, Generality, and Soundness.

- **Coherence**

  - Motivation of Hadamard product (raised by 3mhu)

    The motivation for the Hadamard product stems from addressing computational redundancy in inverted bottlenecks, as evidenced by prior work on feature reuse, and is further justified by its proven efficacy in enhancing model expressivity through implicit high-dimensional mapping.

  - Novelty of sub-modules of ACH (raised by 3mhu)

    The novelty lies not in the sub-components themselves, but in their synergistic integration to enable a learnable cross-Hadamard product, where each component is essential for facilitating the discrete selection process, ensuring gradient flow, and maintaining training stability.

  - Significance of object detection experiments (raised by ZaiK)

    The object detection results demonstrate the ACH module's core strength as a superior backbone component, providing consistent gains under a fixed computational budget; the marginal improvement is attributed to using a standard detection head not optimized for ACH's principles, and the true potential for breaking the efficiency bottleneck lies in the future work of co-designing a specialized neck/head architecture.

  - Important related work discussion  (raised by ZaiK)

    With analyzed the articles provided by the reviewers, we constructs a taxonomy for Hadamard product applications to clearly position the ACH module: while it falls under the category of efficient operators, its introduction of a learnable, cross-channel feature selection mechanism fundamentally distinguishes it from existing works—such as MogaNet's static gating and Manba's spatial recurrence—thereby solidifying its novelty.

- **Theoretics**

  - Mathematical description of ACH (raised by qj3w)

    We provide a more detailed algorithm analysis that combines code implementation and tensor computation flow.

  - Why the use of dysoft (raised by ZaiK)

    The dySoft module is indispensable because it acts as a learnable statistical compression gate that enforces boundedness and smoothness on the statistical mapping of the cross-Hadamard product, thereby suppressing multiplicative variance explosion and ensuring training stability without violating the channel-heterogeneity prior established by BatchNorm.

  - Why the high dimensional priority of ACH (raised by ZaiK)

    The ACH module exhibits a natural dimension preference: it is most effective in high-dimensional layers where channel redundancy is high and cross-channel feature integration is crucial, while standard convolutions remain optimal in low-dimensional layers for capturing fundamental spatial patterns, which aligns with established CNN and Transformer design principles.

- **Generality**

  - On the relation of attention mechanism (e.g. Transformer) (raised by 3mhu)

    With experiments on replacing FFN of Transformer encoder within MobileViTs, ACH module achieves higher accuracy with fewer parameters and FLOPs on the CIFAR-100 dataset.

  - On the interdisciplinary downstream task experiments (e.g. NLP) (raised by qj3w, ZaiK)

    The ACH module demonstrates generality in NLP tasks by achieving performance competitive with a standard, computationally larger FFN while using a significantly compressed architecture.

- **Soundness**

  - Demand of measured compute complexity (raised by ZaiK)

    The computational advantage of the ACH module is empirically validated through extra experiments, resulting in significantly lower FLOPs and MACs, especially at high expansion ratios, as demonstrated across various channel configurations.

  - Demand of hyperparameters decision experiments (raised by ZaiK)

    The hyperparameters for the Gumbel-TopK's temperature modulation were selected as the current optimal set based on empirical tuning, as evidenced by comparative results showing superior performance on CIFAR-100.

We are pleased to confirm that all reviewers have accepted our rebuttal as of November 27th, with one explicitly recommending a score improvement. A revised manuscript incorporating all agreed-upon changes, along with a change log, will be submitted shortly. We sincerely thank the reviewers for their insightful comments, which have significantly strengthened our paper.

---

### Author Response · Authors · 2025-12-03
**Change Log of Latest Revision**

1. Revised the structure of Section 2.1 (Research on Hadamard) and incorporated additional references.
2. Expanded the discussion on object detection experiments.
3. Included replacement experiments with MobileViT in the experimental section of the main text.
4. Added an indispensability analysis of DySoft to the Appendix.
5. Added​ extra computational efficiency measurements to the Appendix.

---

### Meta-Review · Area_Chair_aSn1 · 2026-01-07

**Summary:**

This paper proposes the Adaptive Cross-Hadamard (ACH) modulu by utilizing channel attention-guided feature gating and differentiable discrete sampling along with Hadamard (element wise). The three reviewers pointed out multiple suggestive and critical concerns regarding different aspects of this paper, including but not limited to:

1. The motivation for technical choices is insufficient. The paper lacks a clear and in-depth discussion of the motivation behind its core technical components.
2. The novelty in sub-components is potentially limited. The novelty of some technical components appears to be incremental, such as the section on DIFFERENTIABLE DISCRETE SAMPLING.
3. The experimental scope is limited. The experimental evaluation is confined to CNN-based architectures. In the current deep learning landscape, demonstrating effectiveness on Transformer-based models is crucial for showing broad applicability, especially for a method focused on efficiency. The absence of such experiments makes the generalizability of the proposed Hadaptive-Net unclear.
4. A precise description of the end to end method is suggested to be listed as an algorithm. Equation 10 is somewhat ambiguous. The dysoft operator is somewhat ambiguously applied to the output of the pairwise Hadamard computation.
5. The performance improvement is marginal. For object detection in Tab. 6, the gains of Hadaptive-Net-L over GhostNetV3 or MobileNetV4 are a bit marginal.
6. There is no evidence offered to robustness under distribution/drift, input permutation, adversarial queries, or transfer to other domains (e.g., audio, language).
7. Some important recent and closely related methods are missing.
8. The hyperparameter sensitivity is missing.
9. There are some typos. The presentation needs to be improved.

Most concerns are addressed by the rebuttal, and therefore, I recommend acceptance. I strongly encourage the authors to incorporate the rebuttal into the final version.

**Reviewer Concerns:**

Most concerns are addressed by the rebuttal.

**Reviewer Scores:**

None.

---

### Decision · Program_Chairs · 2026-01-26

Accept (Poster)